
# A Harmonised Instrumental Earthquake Catalogue for Iceland and the Northern Mid-Atlantic Ridge

Kristján Jónasson[1], Bjarni Bessason[2], Ásdís Helgadóttir[1], Páll Einarsson[3], Gunnar B. Guðmundsson[4], Bryndís Brandsdóttir[3], Kristín S. Vogfjörd[4], and Kristín Jónsdóttir[4]

[1]Faculty of Industrial Engineering, Mechanical Engineering and Computer Science, University of Iceland, Hjarðarhagi 2, Reykjavík, Iceland
[2]Faculty of Civil and Environmental Engineering, University of Iceland, Hjarðarhagi 2, Reykjavík, Iceland
[3]Science Institute, University of Iceland, Sturlugata 7, Reykjavík, Iceland
[4]Icelandic Meteorological Office, Bústaðavegur 7−9, Reykjavík, Iceland

**Correspondence:** K. Jónasson (jonasson@hi.is)

**Abstract.** A comprehensive catalogue of historical earthquakes, with accurate epicentres and homogenised magnitudes is a crucial resource for seismic hazard mapping. Here we update and combine catalogues from several sources to compile a catalogue of earthquakes in and near Iceland, in the years 1900–2019. In particular the epicentres are based on local information, whereas the magnitudes are based on teleseismic observations, primarily from international on-line catalogues. The most re-
5 liable epicentre information comes from the catalogue of the Icelandic Meteorological Office, but this is complemented with information from several technical reports, scientific publications, newspaper articles, and modified by some expert judgement. The catalogue contains 1272 $M_W \geq 4$ events and the estimated completeness magnitude is $M_W$ 5.5 in the first years, going down to $M_W$ 4.5 for recent years. The largest magnitude is $M_W$ 7.01. Such melting of local and teleseismic data has not been done before for Icelandic earthquakes, and the result is an earthquake map with no obviously mislocated events. The catalogue
10 also lists additional 5654 earthquakes on the Mid-Atlantic Ridge, north of 43°, with both epicentres and magnitudes determined teleseismically. When moment magnitudes are not available, proxy $M_W$ values are computed with $\chi^2$-regression, normally on $M_S$, but exceptionally on $m_b$. All the presented magnitudes have associated uncertainty estimates. The actual combined seismic moment released in the Icelandic earthquakes is found to be consistent with the moment estimated using a simple plate motion model. The catalogue is named ICEL-NMAR and it is available online at http://dx.doi.org/10.17632/7zh6xg22cv.1.

## 1 Introduction

Seismic hazard in Iceland is the highest in Northern Europe and is comparable to that in Southern Europe. The seismicity is caused by tectonic movements of the plate boundary of the North-America plate and the Euro-Asia plate crossing the island, as well as volcanic activity (Einarsson, 1991, 2008). Based on historical records, faulting mechanisms, and tectonic context, it can be argued that earthquakes larger than about $M_W$ 7.2 are not to be expected (Halldórsson, 1992a). This is further
20 supported by the limited thickness of the seismogenic part of the Icelandic crust, about $8−12$ km (e. g. Stefánsson et al. 1993). Since the settlement of Iceland in the 8th or 9th century A.D. destructive earthquakes have repeatedly been reported in local



chronicles with descriptions of structural damage and fatalities (Sólnes et al., 2013). However, because of the low population density, the losses and number of deaths and injuries has been low and gained little global attention. The main characteristic of the seismicity are shallow ($< 10$ km) strike-slip earthquakes as well as earthquakes related to volcanic activity. The first instrumentally recorded earthquakes in Iceland occurred in 1896 when six destructive earthquakes struck in South Iceland in a two week period (Ambraseys and Sigbjörnsson, 2000; Sigbjörnsson and Rupakhety, 2014). These events where recorded at several stations in Europe: England, France, Poland and Italy, equipped with rather primitive seismographs (Sólnes et al., 2013, p. 579−583). Damped seismographs, which could measure absolute ground motion, were introduced around the year 1900, allowing (later) magnitude computation. The first seismograph was installed in Iceland in 1909 and was operated until 1914, and again from 1925 when continuous operation was secured.

The main motivation behind this study is to construct a harmonised earthquake catalogue for Iceland to use in seismic hazard analysis. A selection criterion for inclusion is that the earthquake was instrumentally recorded by seismic centres outside Iceland and assigned an $M_S$, $m_b$ or $M_W$ value, and that it is listed either in the International Seismological Centre (ISC) Bulletin Event Catalog (ISC, 2020), or in the catalogue of Ambraseys and Sigbjörnsson (2000), which lists and reappraises internationally recorded earthquakes in the region $62°−68°$N and $12°−26°$W (Fig. 1), in the period $1896−1995$. This catalogue will be referred to as the AMB-SIG catalogue. The new catalogue contains reappraised magnitudes and locations for earthquakes in the AMB-SIG region (referred to as ICEL) and the period $1900−2019$, a total of 1272 earthquakes.

The magnitudes are all copied or computed from ISC, AMB-SIG, or the Global Centroid Moment Tensor (GCMT) catalog (GCMT, 2020). $M_W$ values are provided for all earthquakes. They are of three types: (a) taken directly from the GCMT catalogue if available there (the golden standard), (b) averaged or copied from values in the ISC catalogue, or (c) proxy values computed with regression using $M_S$ or $m_b$. For the regression, region-specific magnitude relationships were developed using data from a larger region, referred to as NMAR. This region follows the Northern Mid-Atlantic Ridge (Fig. 2), and includes all of the region AOI (Atlantic Ocean and Iceland) of Grünthal and Wahlström (2012). A byproduct of our study is therefore a catalogue of 6926 events in the whole NMAR region (including the 1272 ICEL events). Locations of events outside ICEL are copied directly from the ISC catalogue, and magnitudes are obtained in the same way as inside it. The magnitude range of the new catalogue is $M_W$ $4−7.08$, as events $M_W < 4$ were omitted.

Opposite to magnitudes, earthquake locations in the international catalogues are often very inaccurate (by tens of km), being based on teleseismic data. One of the innovations in the new catalogue is therefore to use local information on epicentres. The primary sources for these locations are catalogues compiled at the Icelandic Meteorological Office (IMO), seismological bulletins, newsletters and reports published by the IMO and the University of Iceland Science Institute (UISI), journal articles with results of studies on Icelandic earthquakes, and contemporary accounts of earthquakes from newspapers. These sources are complemented by the authors' judgement.

All origin times are copied from the international catalogues, but, since origin times after 1990 are probably more accurate in local catalogues, the new catalogue also reports these.

An early published list of instrumentally recorded earthquakes in Iceland and the surrounding oceans appeared in Gutenberg and Richter's book (1949), p. 196, 207, which lists 60 large earthquakes in the period $1910−1945$ in the NMAR region, of

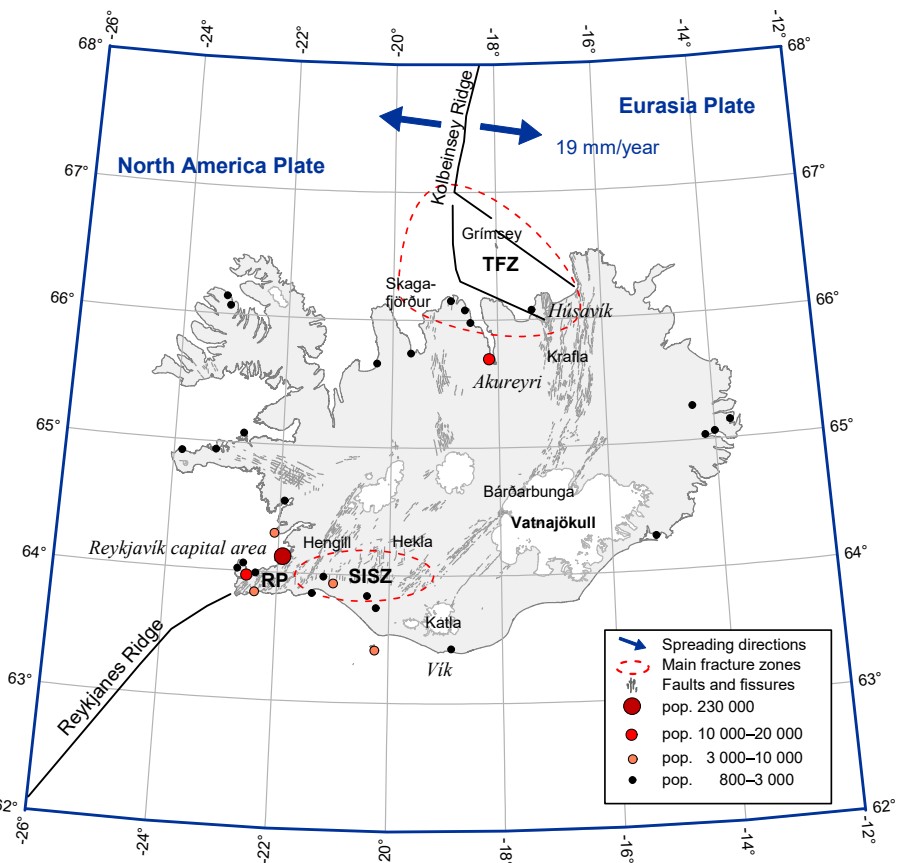

**Figure 1.** The ICEL region, $62°-68°$N and $12°-26°$W. The figure shows place names in Iceland mentioned in the article. Towns and villages with 2020 population of at least 800 are also indicated as well as the Tjörnes Fracture Zone (TFZ), the South Iceland Seismic Zone (SISZ), and the Reykjanes Peninsula (RP)

these 8 are in the ICEL region. Six years later Tryggvason (1955) compiled a list of earthquakes $M \geq 5\frac{1}{4}$ in $1927-1945$, 121 in NMAR, of these 22 are in ICEL.

Since shortly after the IMO was established, it has been responsible for monitoring earthquakes in Iceland. From the begin-
ning, accounts of earthquakes have been published in the IMO monthly newsletter *Veðráttan* (the Weather) (IMO, 1924–2006), in addition the Seismological Bulletin (IMO, 1926–1973) was compiled and distributed to seismological centres abroad, and since 1975 computerised earthquake catalogues have been kept, and made available to scientists working elsewhere. After 1965 earthquake research took off at the University of Iceland, and has flourished ever since with a number of case studies, as well as historical summaries.

The new century has seen a surge in the publication of local and global earthquake catalogues, and Iceland is not an exception. The aforementioned catalogue of Ambraseys and Sigbjörnsson (2000), covers the same ICEL region as the current study and lists 415 earthquakes with $M_S$ and/or $m_b$ magnitudes. The epicentres for a portion of these were reassessed, but for the

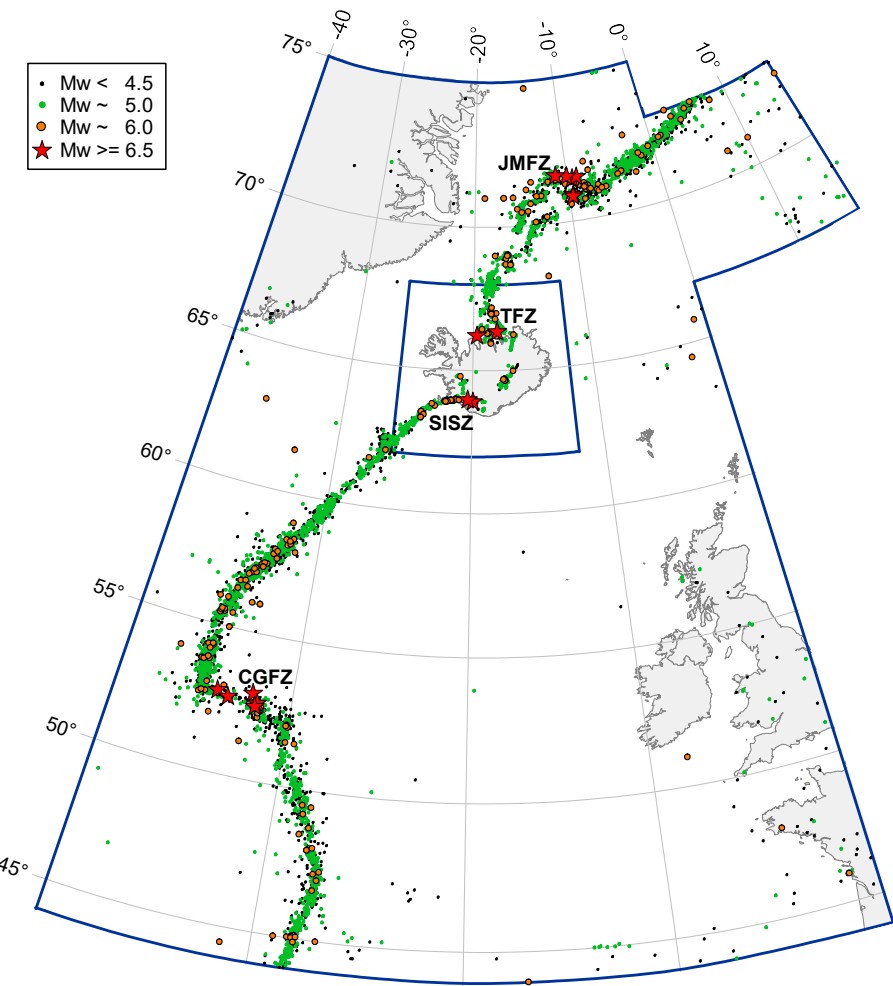

**Figure 2.** The NMAR region, $44°-75°$N, $0°-40°$W, and $67°-73°$N, $0°-17°$E. The small part in the eastern hemisphere is added to make the region include all of the AOI region of Grünthal and Wahlström (2012). The ICEL region is also marked in. Four main seismic zones are marked on the map, i. e. Charlie-Gibbs Seismic Zone (CGSZ), South-Iceland Seismic Zone (SISZ), Tjörnes Fracture Zone (TFZ), and Jan Mayen Fracture Zone (JMFZ).

remaining ones, inaccurate teleseismically determined locations were given. To our knowledge, this is the only catalogue apart from the current one where local locations and global magnitudes have been combined. Unfortunately this catalogue was only

published in a very limited distribution, and it is not available online.

Grünthal and Wahlström (Grünthal and Wahlström, 2003) compiled a historical catalogue of earthquakes in Central and Northern Europe until 1993, with magnitudes and locations in Iceland taken from a data file obtained from the IMO. These data were compiled at the IMO independently of the IMO catalogue discussed in Sect. 2.2.1, and are still available on the IMO website (hraun.vedur.is/ja/ymislegt/storskjalf.html). The locations are reasonably accurate, but the resulting $M_W$ magnitudes





are exaggerated, often by a whole magnitude (less for the most recent earthquakes, or $\sim 0.2-0.3$ magnitudes). The work
on this catalogue continued with a number of subsequent projects (Grünthal et al., 2009; Grünthal and Wahlström, 2012;
Grünthal et al., 2013), under several acronyms, CENEC (CEntral, Northern and northwestern European Catalogue), EMEC
(European Mediterranean earthquake catalogue), SHARE (seismic hazard harmonization), and SHEEC (SHARE European
earthquake catalogue). For the Iceland region, all these projects adopt the original 2003 catalogue, adding data (locations and
local magnitudes) after 1990 from IMO's catalogue. Among the products of these studies were hazard maps for Iceland where
the hazard was greatly overestimated in many places, among them in the Reykjavík capital area (Woessner et al., 2015).

In 2010 the ISC initiated work on a global catalogue of large earthquakes since 1900, ISC-GEM (ISC-Global Earthquake
Model). The first version was released in 2013 and the work is ongoing, with version 6 being released in 2019. The magnitude
thresholds are: $1900-1917$: $M_S \geq 7.5$, $1918-1959$: $M_S \geq 6.25$, $1960-2015$: $M_S \geq 5.5$ (Storchak et al., 2013; Di Giacomo
et al., 2015). The catalogue contains 40 earthquakes in the ICEL region.

Panzera et al. (2016) compiled a catalogue of earthquakes in South-Iceland $1991-2013$. It reports locations and magnitudes
from IMO's database, cleaned and corrected, as well as proxy $M_W$-values based on regression of GCMT-magnitudes on the
IMO data, like the CENEC/EMEC catalogues. It has more than $150\,000$ events with magnitudes down to $M = 0$.

## 2   Sources and data

This section discusses the primary sources used to compile the new ICEL-NMAR catalogue. These sources consist of four
international catalogues, used primarily to obtain and/or compute magnitudes, and several types of local Icelandic sources
used as a basis for event locations. The local sources include the catalogue of the IMO, scientific publications, seismological
bulletins, newsletters and technical reports, as well as newspaper articles. The section concludes with a few remarks on how
individual events in different sources have been matched up.

### 2.1   International catalogues

### 2.1.1   The ISC Bulletin Event Catalog

The ISC database (2020) contains data on earthquake location and magnitude contributed by several seismological agencies
from around the world. For each earthquake a single origin time (UTC) and location with multiple magnitude values are
provided. The magnitudes are of several different types, but in the present work only $M_S$, $m_b$ and $M_W$ are considered.
Magnitudes coded as $m_S$ and $M_s$ are treated as $M_S$, and similarly for varying capitalization of $m_b$. In addition in the period
$1955-1970$ there are a few magnitude values marked as $M$ and these are also treated as $M_S$ cf. (Sykes, 1965). When both
$M$ and $M_S$ values are available for an earthquake the difference is small. Each magnitude is either marked ISC, to signify
that the value is computed by ISC themselves, or else it is marked with the abbreviation of a submitting agency. The ISC-
marked values are referred to as reviewed, and according to Storchak et al. (2017), "seismic events are reprocessed resulting in
more robust and reliable mb and MS magnitudes". Di Giacomo and Storchak (2016) say that ISC puts considerable effort into





relocating earthquakes and recomputing their magnitudes. They also recommend that preference be given to three agencies, CTBTO (Comprensive nuclear-Test-Ban Treaty Organization, also known as International Data Centre, IDC, Vienna), MOS (Geophysical Survey of Russian Academy of Sciences, Moscow), and USGS (United States Geological Survey).

### 2.1.2  The GCMT Earthquake Catalog

The GCMT catalogue (2020) contains data on seismic moment tensors with associated $M_W$ magnitudes of large earthquakes ($M_W \geq 5$) around the world, starting in 1976 (Dziewonski et al., 1981; Ekström et al., 2012). This is considered to be the most authoritative catalogue providing $M_W$ (Di Giacomo and Storchak, 2016). There are 653 events in the NMAR region in this catalogue, and all but 9 of them are also in the ISC catalog, marked as originating from GCMT. In 482 cases the $M_W$ match but in 171 cases there is a mismatch of 0.1 magnitude, and the average is used here.

### 115  2.1.3  The catalogue of Ambraseys and Sigbjörnsson

Ambraseys and Sigbjörnsson (2000) published an earthquake catalogue for Iceland or more specifically for the region shown in Fig. 1. The catalogue covers exactly one century, i. e. from 1896 to 1995, and lists 422 earthquakes. The catalogue is based on teleseismic data from seismological bulletins, and information from books, journals, newspapers and reports. The authors recalculated surface magnitudes ($M_S$) and locations when possible. Ambraseys and Sigbjörnsson (2000) mention that the

greatest outstanding problem was the epicentral accuracy, particularity for pre-1960 macroseismic and instrumental events. They specially remark that epicentres before 1918 reported by the British Association for Advancement of Science (BAAS) are crude, as well as epicentres estimated by the ISC before 1950, although to lesser degree (Ambraseys and Sigbjörnsson, 2000). This catalogue contains valuable information for the time period from 1900 to 1960 when fewer records are available from other catalogues.

### 125  2.1.4  The USGS Earthquake Catalog

The USGS catalogue (2020) provides one magnitude value per earthquake ($M_W$, $M_S$ or $m_b$), which is in almost all cases identical to the corresponding USGS-labeled value in the ISC database. However the locations in the USGS catalog are different from those in the ISC catalog, the difference frequently amounting to a few tens of kilometers.

### 2.2  Local sources and catalogues

### 130  2.2.1  The catalogue of the Icelandic Meteorological Office

The Icelandic Metorological Office (IMO) in Reykjavík has been responsible for monitoring earthquakes in Iceland since shortly after its foundation in 1920 when the Mainka seismograph mentioned in the introduction was reinstalled there in 1925. A second Mainka instrument was installed in 1927, also in Reykjavík. Data processing was conducted at the IMO and the results were published in Seismological Bulletins (IMO, 1926–1973) which were sent to several seismological agencies

around the world. These results were mainly phase readings and reports of felt earthquakes along with a few locations.



After 1980 the IMO reanalyzed these data and combined them with other local and global sources, e. g. the University of Iceland (UI) reports discussed in the next subsection, and Kárník (1968), to mention a few. The resulting event locations and magnitudes form the basis of IMO's catalogue for the period 1926−1952.

In 1951-1952 three Sprengnether short-period seismographs, measuring all three components of motion, were installed in
Reykjavík and the old seismographs were moved to Akureyri in North Iceland and to Vík in South Iceland (Fig. 1), and in the following two decades several more instruments were installed.

As detailed in the next subsection, the University of Iceland Science Institute (UISI) initiated several research projects involving seismic measurements after 1970. Many of these were in cooperation with the IMO, and at the same time IMO's network continued to expand. As before the resulting data were published in the Seismological Bulletins. The IMO catalogue
1952−1974 is based on these and a digital-only bullettin for 1974.

From 1975 to 1986 no bulletins were published, and to fill up this gap, phase readings from the UISI and the IMO stations were merged and reanalyzed to compute locations and magnitudes. This work was carried out at the IMO after 1990, and earthquakes of magnitude $M_L > 3$ were entered into the IMO database. The database for this period is somewhat preliminary and incomplete, as manual review is lacking. The period 1987−1990 is also in the IMO database, with results based on
*Mánaðaryfirlit jarðskjálfta* (Monthly reports of earthquakes) (IMO, 1987–1990), published by the IMO in cooperation with the UISI.

In 1991 a digital seismic system, the South Iceland Lowland (SIL) system was implemented by the IMO (Stefánsson et al., 1993; Bödvarsson et al., 1996). As the name implies, it began in South Iceland, but was gradually expanded to cover all geologically active areas in the country. In 2020 around 80 stations are in operation in the SIL-network. Even if the system did
not cover the whole island to begin with, all events of magnitude $M_L > 4$ occurring within a few tens of km offshore should be present for the whole period. Locations and local magnitudes are automatically computed by the system, all automatically located events are manually reviewed, and the location recomputed. The IMO catalogue from 1991 is based on the SIL system analysis.

### 2.2.2 Data from the University of Iceland Science Institute

Research on historical seismicity at the University of Iceland relies heavily on reports by Tryggvason (1978a,b, 1979) and Ottósson (1980). Tryggvason's reports are based on the early seismographic observations at IMO and overseas for the years 1930–1960, augmented by felt reports and newspaper reports. Ottósson's report on earthquakes during 1900–1930 is based on felt reports and newspapers, supported by rare teleseismic observations.

Technical advances and increasing interest in crustal activity following the Surtsey eruptions in 1963–1967 led to a pro-
liferation of seismic observations in Iceland in the late 1960ies (Einarsson, 2018). Cooperation started between the UISI and the Lamont-Doherty Earth Observatory (LDEO) at Columbia University in NY. A team from LDEO came to Iceland with several portable seismographs to study the background seismicity of the mid-Atlantic plate boundary (Ward, 1971). A network of six stations was operated on the Reykjanes Peninsula segment of the boundary during 1971–1976 (Björnsson et al., 2020), augmented by a dense network in the summers of 1971 and 1972 (Klein et al., 1973, 1977). The work continued by building an





island-wide network of short-period, vertical component seismographs, designed and built at UISI. The installation began in South Iceland in 1973 and the network was gradually expanded in the following years, to the Tjörnes Fracture Zone (TFZ) in North Iceland in 1974, and to other parts in 1975–1979. A telemetered network was installed in Central Iceland in 1985. These networks provided valuable data on major events such as the Krafla volcano-tectonic episode of 1975–1984 (Einarsson and Brandsdóttir, 1980; Brandsdóttir and Einarsson, 1979; Buck et al., 2006; Wright et al., 2012), the Hekla eruptions of 1980 and 1991 (Grönvold et al., 1983; Soosalu and Einarsson, 2002) and the Gjálp eruption in Central Iceland in 1996 (Einarsson et al., 1997), as well as the location of the major seismically active structures of Iceland (Einarsson, 1991). After 1991, the analog seismic stations were gradually replaced by the SIL-system discussed in the previous subsection. The last analog stations were dismantled in Central Iceland in 2010. Some of the data gathered by the seismic network discussed above, including epicentres, are documented in the *Skjálftabréf* (Earthquake letter) (UISI, 1975–1988).

### 2.2.3 Newspapers

Newspapers are an important source on earthquakes in Iceland during the first part of the 20th century. The web page http://timarit.is provides search access to all newspapers published in Iceland during $1830-2016$. News about earthquakes often provide direct or indirect information on their epicentres. In the current work we have used this data source extensively to check the correctness of the sources listed in the previous sections, and when deemed appropriate, to correct earthquake locations for the new catalogue.

## 2.3 Combining catalogues

All the catalogues, that need to be combined for the current study, have their own version of both origin time and location of each earthquake. As proposed by Jones et al. (2000) and several later publications we consider two records that differ by less than 16 s and 100 km to refer to the same earthquake. In a few cases we have found that this window is a little too narrow and we have made an appropriate manual adjustments. Furthermore, the AMB-SIG catalogue only provides times to the nearest whole minute, so for that a 90 s time window is used. For each earthquake, the ISC-time, all available locations (ISC, AMB-SIG, IMO, other local sources), and all available magnitude values of different types ($M_W$, $M_S$, $m_b$) and from different catalogues/contributors are entered into a data file. This file is then used for further processing as described below.

## 3 Earthquake locations

When accurate instrumentally determined location of an earthquake is missing, which applies to a large part of the study period, several methods may be used to determine the epicentre. Sometimes the historical accounts, discussed in Sect. 2.2 provide quite accurate locations, especially in inhabited areas. For the past decades a major effort has been devoted to the mapping of surface expressions of earthquake faults in Iceland, and these often indicate the location of historical earthquakes (Einarsson, 2015). Furthermore, the main faults tend to produce microearthquakes detected with the SIL network. By relative locations, detailed maps of the subsurface faults can be produced(Slunga et al., 1995). Combining all these methods and adding expert judgement



will normally give a much more accurate locations than those provided by the international catalogues, and the same holds for many of the locations in the IMO catalogues, even before 1990.

The remainder of this section describes details of how this methodology has been applied for several subperiods of the study period.

## 3.1 The period until 1990

In the period 1900−1925 there are 22 earthquakes in the ICEL region listed in our data file. All of these are in the AMB-SIG catalogue, and 4 are also in the ISC catalogue, originally coming from Gutenberg and Richter (1949). The authors have viewed all these earthquakes on a map, checked newspapers articles for contemporary accounts of them (using the web service timarit.is mentioned in Sect. 2.2.3), as well as scientific publications, in particular the report of Ottósson (1980). The result of this scrutiny is to use the AMB-SIG location for 14 earthquakes, the aforementioned report for one event, and relocate 6 events using the methodology described at the beginning of this section. In the new catalogue these location sources have been specified as "Amb-Sig", "Report" and "New" respectively. Finally, for the 22 January 1910 earthquake we use the location provided by (Stefánsson et al., 2008), 20 km offshore North-Iceland. This source is marked as [1] in the catalogue, with details in an accompanying reference list.

In the period 1926−1955 there are 98 earthquakes in our data set, and their location has been scrutinised in the same way. Sometimes we can take into account that an origin time is within a known earthquake series. For this period additional data sources are the IMO catalogue (Sect. 2.2.1), as well as the reports of Tryggvason (1978a,b, 1979) which often provide direct epicentres. This results in using 36 AMB-SIG locations, 21 IMO locations (marked "IMetO" in the new catalogue), 34 locations from the reports, 4 computed as average of the most believable reported locations (marked "Average"), and 3 relocated (marked "New").

In the period 1956−1990 there are 380 earthquakes in the data file. Having multiple local seismometers opens the possibility of computing locations from local measurements. Such locations have found their way into several of our sources, but the quality is variable. There are several journal articles stemming from this period providing locations for 41 earthquakes and our choice is to trust these. The relevant articles are listed in the reference list in the readme-file accompanying the catalogue, and specified as [2], [3], etc. in the catalogue itself. Some of the articles are also cited in Sect. 2.2.2 above. Available locations for the remaining 338 earthquakes were viewed on a map, upto 4 locations per earthquake: From AMB-SIG, IMO, ISC, and one of the earthquake reports, newsletters or bulletins. It transpired that none of these sources could be used as an overall first choice, but instead we had to select the most believable one in each case, or sometimes take an average or relocate. The result was to use AMB-SIG for 59 cases, the IMO catalogue for 107, ISC for 36, 12 from reports, 55 locations from the *Skjálftabréf* (Earthquake letter) (UISI, 1975–1988) (marked "Letter"), 14 averaged, and 56 relocated.

## 3.2 Earthquakes after 1990

For the period 1991−2019 our data file contains 980 earthquakes in the ICEL region. With the introduction of the SIL system described in Sect. 2.2.1, the quality of the local epicentre information vastly improved after 1990. We have viewed maps of





these locations together with ISC and USGS locations, along with a background layer showing microearthquake activity. From

this comparison it was evident that the errors in the teleseismic locations are in many cases tens of kilometers (c. f. Sect. 3.3), whereas the SIL locations are very convincing, normally accurate to a few km (1 or 2 inside the network, but somewhat more outside). The only region where the SIL-locations seem suspect is on the Reykjanes Ridge, more than 150 km offshore, or approximately south of 63°N. This inaccuracy is not important for future work with these data e. g. in hazard analysis, and we have chosen to use the ISC locations for the relevant 40 earthquakes. In addition there are 33 ISC-earthquakes in the

ISC catalogue missing from the SIL catalogue. Of these, 25 were located far offshore and 8 were in or near the Bárðarbunga caldera, in the uninhabited interior of Iceland. The earthquakes near the caldera were relocated to the caldera itself, and the ISC locations for the offshore events were retained.

### 3.3 Accuracy of earthquake locations

To get some indication of the accuracy of event locations in the international catalogues the locations in the AMB-SIG and

the ISC catalogues have been compared. For 292 events in both catalogues (period 1910−1996), the maximum mismatch in location is 113 km, the median is 10.0 km, and in 90% of cases the difference is $< 30$ km. The accuracy does not seem to increase markedly with time or with earthquake magnitude. A similar comparison between the ISC and the USGS catalogues (1973−2019) gave a maximum difference of 108 km and median difference of 9.5 km. Comparison of ISC and SIL in the ICEL region (925 events; 1991−2019) gave a median of 5.7 km with 93% $< 30$ km, and ISC-USGS comparison in the ICEL

region (630 events; 1973−2019) gave a median of 15.3 km with 89% $< 30$ km.

## 4 Earthquake sizes

Contrary to earthquake locations, where local information is better, estimating earthquake sizes with teleseismic data is often easier and more reliable than using regional and local data. The dominant periods at teleseismic distances are longer and the structure is smoother, and therefore the waveforms fit better (Wang et al., 2009; Karimiparidari et al., 2013; Yadav et al., 2009).

Modern earthquake catalogues generally provide moment magnitudes for all earthquakes larger than about $M_W$ 4. For earthquakes, whose source mechanism and magnitude have not been modeled by moment tensor inversion of seismic data, regression on surface or body-wave magnitudes is customarily used to obtain proxy $M_W$ values, and this procedure is followed here. As mentioned in the introduction a larger collection of earthquakes than is really needed in the Iceland context is used to construct the $M_S$-$M_W$ and $m_b$-$M_W$ regression relationships, thus killing two birds with one stone, improving the accuracy of

these relationships, and getting a larger catalogue of 6926 earthquakes. The data file discussed in Sect. 2.3 above contains some earthquakes that are to small to be included in the catalogue, but are used in the regression in order to improve the relationship for small magnitudes.

For each earthquake there are usually several $m_b$-values, contributed by different agencies, and the same applies to $M_S$, and sometimes also $M_W$. These values must be apropriately averaged or selected before they can be used in the regression.

This subtask is dealt with in the next subsection, followed by a subsection on uncertainty in the magnitude estimates in the





context of previous studies. Subsection 4.3 discusses the proxy regression, and finally there are two short subsections on the uncertainty in the proxy and local magnitudes.

## 4.1 Best estimates of $M_W$, $M_S$ and $m_b$

### 4.1.1 Estimates of $M_W$

In the NMAR region 873 earthquakes in our data have modeled moment magnitudes, of these 147 are in the ICEL region. The GCMT catalogue is the golden standard for moment magnitudes, and available GCMT $M_W$ values are used verbatim, 666 in total in the larger NMAR region. The magnitudes range from $M_W$ 4.51 to 7.08, stemming from the period 1976−2019. Additional 208 earthquakes have $M_W$-values from other sources, 204 are from the Swiss Seismological Service (the "Zurich Moment Tensors" (ZUR-RMT)), all stemming from the period 2000−2005, and 3 are from the USGS catalogue. In addition
to the 204 earthquakes, 61 earthquakes are listed in both the GCMT and the ZUR-RMT catalogues, with ZUR-RMT values on average 0.08 magnitudes higher (standard deviation 0.09). The common values are in the range 4.8−6.6 and a graph of $M_{\text{GCMT}}$ against $M_{\text{ZUR-RMT}}$ shows that the relationship is approximately linear with slope 1, which justifies using $-0.08$ as an agency correction for ZUR-RMT. More precisely, we set $M_{\text{est}} = M_{\text{ZUR-RMT}} - 0.08$, and the estimated values are in the range 3.62−5.22.

Similarly GCMT and USGS have 109 common events, with a correction of 0.00 and standard deviation of 0.08, and we set $M_{est} = M_{\text{USGS}}$ for the 3 events. Other agencies which provide 35 additional $M_W$ values in the ISC catalogue have been compared with the GCMT catalogue in the same way, but in all cases the standard deviation is too high to include them.

### 4.1.2 Estimates of $M_S$

The data contains 5076 $M_S$ values for earthquakes in the NMAR region, of these 1074 in the ICEL region. This time the
golden standard consists of reviewed values in the ISC catalogue. The situation is somewhat complicated by the fact that three important sources for magnitudes in the first half of the catalogue period have very little overlap with these reviewed values, so that corresponding agency corrections cannot be determined. In fact all sources have small overlap with ISC before 1965. The period has therefore been divided in two, 1900−1964, and 1965−2019.

Of the 317 $M_S$ values before 1965, 43 are ISC-reviewed. The remaining 274 $M_S$ values come from a total of 24 other
sources, the most important being Ambraseys and Sigbjörnsson (2000), Sykes (1965) (PAL in the ISC catalogue), and the California Institute of Technology in Pasadena (PAS). For each of these earthquakes a direct average of available magnitudes is used.

Of the 4759 earthquakes occurring since 1965, 2828 have ISC-reviewed magnitudes, again used unchanged. The remaining 1931 events have $M_S$ values from a total of 33 sources. After pooling agencies with fewer than 20 events all sources have
sufficient overlap with ISC to estimate an agency correction, $\Delta_i$, computed as the average of all available differences, $\delta_i = M_{\text{ISC}} - M_i$, where $M_i$ is the magnitude estimated by agency $i$. When only one source is available, $M_{\text{est}}$ is set to $M_i + \Delta_i$, but




otherwise a weighted average is computed using

$$M_{\text{est}} = \sum_i w_i(M_i + \Delta_i), \tag{1}$$

where the $w_i$ are normalised weights, and the sum is taken over all available $M_i$. If the $\Delta_i$ are independent it is optimal to

weigh with their inverse variance, and, even if not optimal, it is more robust to use the same weights when the $\Delta_i$ are correlated (Schmelling, 1995). To be precise, $w_i = (1/\sigma_i^2)/\sum_i(1/\sigma_i^2)$, where $\sigma_i$ is the standard deviation of the available $\delta_i$. The lowest corrections $(0.02-0.04)$ and the lowest standard deviations $(0.10-0.16)$ are those for AMB-SIG, CTBTO, MOS and USGS. Of the 1931 events without reviewed ISC magnitudes, 1802 are contributed by a single agency (the majority, 1373, from CTBTO), and for 129 of them Eq. (1) is used.

### 305 4.1.3 Estimates of $m_b$

Our data file contains 7794 NMAR events with an $m_b$ value, of these 1308 ICEL events. Again it is beneficial to split the period at year 1965. ISC-reviewed values are once more used when available, for 38 earthquakes out of 64 before 1965 and for 5774 out of 7730 since 1965. Of the 26 remaining earthquakes in the first period Ambraseys and Sigbjörnsson (2000) provide $m_b$ for 21 events and USGS provides the last 5. Of the 1892 remaining earthquakes in the second period there are 44 contributors

of $m_b$ values, the largest being CTBTO and USGS. Final $m_b$ values are computed as for $M_S$: 1688 have a single contributor and 268 use Eq. (1). Agency corrections and standard deviations are somewhat higher than for $M_S$, typically $0.1-0.2$ and $0.15-0.25$, respectively.

### 4.2 Uncertainty of magnitude estimates

### 4.2.1 A short survey of uncertainty estimates

Helffrich (1997) discusses the uncertainty of moment magnitudes in the GCMT and USGS catalogues, and his conclusion corresponds to a standard deviation in $M_W$ of 0.05, 0.04, and 0.10, for deep, intermediate, and shallow events, respectively. Kagan (2003) studies the accuracy of earthquake catalogues extensively. Among his conclusions are the standard deviation of $M_W$ for both the GCMT and USGS catalogues on the order of $0.05-0.09$ for deep to shallow earthquakes, $0.07-0.11$ for $M_W$ 6 to 8, and decreasing from 0.11 to 0.06 in the period $1980-2002$. Werner (2008) models the magnitude accuracy of

25000 events during $1980-2006$ with a Laplace-distribution. The confidence interval presented in the article corresponds to the confidence interval of a normal distribution with $\sigma = 0.08$. Finally, Gasperini et al. (2012) conclude with an even lower value, $\sigma(M_W) = 0.07$. Many of the estimates cited above are obtained by dividing the standard deviation of magnitude difference between the USGS and the GCMT catalogues by $\sqrt{2}$, on the assumption that the errors in them are independent and have the same variance. In reality the errors are probably correlated, so that the cited values may be underestimates of the actual

uncertainties.

With a little handwaving Kagan (2003) estimates the uncertainty of $M_S$ in the ISC catalogue to be about 0.2, and that of $m_b$ to be about 0.25. In line with these numbers, Kagan also concludes that when $M_S$ and/or $m_b$ is turned into proxy $M_W$, the





uncertainty is about $3-4$ times higher than when $M_W$ is found with moment tensor modeling. This reckoning is supported by both Werner (2008) and Gasperini et al. (2013).

### 4.2.2 Uncertainty of the best estimates

For earthquakes occurring before 1965, there is not enough data to compute the uncertainty objectively, so that a subjective estimate must be used: For this period the uncertainty in $M_S$ has been set to 0.25, and that in $m_b$ to 0.30.

After 1964, Eq. (1) is used. Let $M$ denote the actual magnitude of an earthquake, and $M_g$ its "golden standard" estimated magnitude (which may be unavailable), $M_{GCMT}$ for moment magnitude and $M_{ISC}$ for the other two magnitudes. Also, let $d = M_g - M$. The uncertainty in $M_g$, or standard deviation of $d$, is set to

$$
\sigma_d = \begin{cases}
0.09 \text{ for moment magnitude} \\
0.18 \text{ for surface magnitude} \\
0.23 \text{ for body-wave magnitude}
\end{cases}
\tag{2}
$$

and these numbers are used directly when $M_g$ is available and $M_{est} = M_g$. Keeping in mind that almost all the earthquakes in the NMAR region are shallow, these uncertainties are perhaps somewhat lower than those quoted in Sect. 4.2.1. However, the accuracy of the global catalogues has probably improved since the quoted studies were carried out, and, furthermore, these studies do not explicitly specify GCMT or reviewed ISC magnitudes.

When $M_g$ is not available, and $M_{est}$ is computed via Eq. (1) the error in the magnitude estimate may be partitioned into several terms:

$$
\begin{aligned}
M_{est} - M &= (M_{est} - M_g) + (M_g - M) \\
&= \sum w_i (M_i + \Delta_i - M_g) + d \\
&= \sum w_i (\Delta_i - \delta_i) + d
\end{aligned}
$$

using that the $w_i$ sum to 1. Treating $d$ and the $\delta_i$ as random variables, and the $\Delta_i$ as constants this gives,

$$
\mathrm{Var}(M_{est} - M) = \sigma_d^2 + \sum_i w_i^2 \, \mathrm{Var}\, \delta_i + 2 \sum_{i<j} w_i w_j \, \mathrm{Cov}(\delta_i, \delta_j) - 2 \sum_i w_i \, \mathrm{Cov}(d, \delta_i)
$$

The first term is given by Eq. (2), and $\mathrm{Var}\, \delta_i$ and $\mathrm{Cov}(\delta_i, \delta_j)$ can be approximated by $\sigma_i^2$ and $\sigma_{ij}$, the data covariance of the available pairs $(\delta_i, \delta_j)$. Finally, for the last term, we have

$$
w_i \, \mathrm{Cov}(d, \delta_i) = r_i \sigma_d \sigma_i
\tag{3}
$$

where $r_i$ is the correlation between $d$ and $\delta_i$. A reasonable constraint is that this correlation is positive: If $M_g$ overestimates $M$, why should $M_i$ overestimate $M$ even more? Another constraint is that the estimated variance in $M_{est}$ is not smaller than when the golden standard $M_g$ can be used. The second constraint corresponds to $r_i = \sigma_i / (2\sigma_d)$. Selecting the middle road with $r_i = \sigma_i / (4\sigma_d)$ seems reasonable: it gives $r_i$ in the range $0.11-0.64$; on average 0.28. This choice corresponds to approxmating





the last term with $\sum_i w_i^2 \sigma_i^2$, and the uncertainty estimate:

$$\text{SD}(M_{\text{est}} - M) = \sqrt{\sigma_d^2 - \frac{1}{2}\sum_i w_i^2\sigma_i^2 + 2\sum_{i<j} w_i w_j \sigma ij} \qquad (4)$$

The root-mean-square (RMS) average uncertainty for all cases where Eq. (1) is used to estimate $M_W$ is 0.113, for $M_S$ it is 0.205, and for $m_b$ 0.302.

### 4.3 Proxy values for $M_W$

In the New Manual of Seismological Observatory Pratice, Bormann et al. (2013) recommend the use of general orthogonal regression to convert between magnitude types when uncertainties in the types differ significantly, as when estimating $M_W$ from $M_S$ or $m_b$. They also recommend using a nonlinear relationship. An implementation of such a procedure is given by Gasperini et al. (2013) which is based on Stromeyer et al. (2004), and we have chosen to follow this procedure. A proxy $M_W$ value is computed from $M_S$ using

$$M_W^{\text{proxy}} = \exp(a + bM_S) + c, \qquad (5)$$

where $M_S$ is the best estimate of Sect. 4.1, $a$, $b$ and $c$ are parameters determined by $\chi^2$-regression using Matlab's optimization toolbox and the formulae in Appendix B of Gasperini et al. (2013) (note that the two terms in curly braces in Eq. (B2) in the Appendix should be squared).

Bormann et al. (2013) also recommend weighing data points in magnitude ranges with low data frequency higher (histogram equalization). We use a moderately weighted regression of this type: an earthquake with moment and surface magnitudes $M_W$ and $M_S$ gets a weight of $M_W + M_S - 2$. The effect is that the largest earthquakes weigh about twice as much as the smallest ones.

There is freedom in the regression to fix one of the uncertainties, $\sigma(M_S)$ or $\sigma(M_W)$, and it is also possible to fix their ratio. If the ratio is taken as 2.0, as in Gasperini's article, the NMAR data gives $\sigma(M_S) = 0.176$ and $\sigma(M_W) = 0.0881$.

Exactly the same method could be used to compute $M_W$ from best estimates of $m_b$. However the NMAR dataset contains much fewer large earthquakes than the one used by Gasperini et al., so when this is attempted, the relationship turns out to be very slightly concave rather than convex (logarithmic rather than exponential). The nonlinearness is so slight that it can be ignored with a linear model. For earthquakes larger than about $m_b = 5.75$ an $M_S$ value is almost always available, and, as explained below, preferred. Thus a model valid for $m_b < 5.75$ is constructed and used:

$$M_W^{\text{proxy}} = a + bm_b, \qquad (6)$$

Earthquakes in the Bárðarbunga caldera (Fig. 1) exhibit a different relationship between $M_W$ and $m_b$ than the rest of the data set: for the same $M_W$, their $m_b$ is ~0.15 higher. Therefore a separate model is used for these earthquakes. The relationship between $M_W$ and $M_S$ is also slightly different in the caldera than elswhere, and for consistency separate models are also used in this case. The ratio used by Gasperini et al., $\sigma(m_b)/\sigma(M_W) = 2.5$, gives $\sigma(m_b) = 0.225$ and $\sigma(M_W) = 0.0900$.


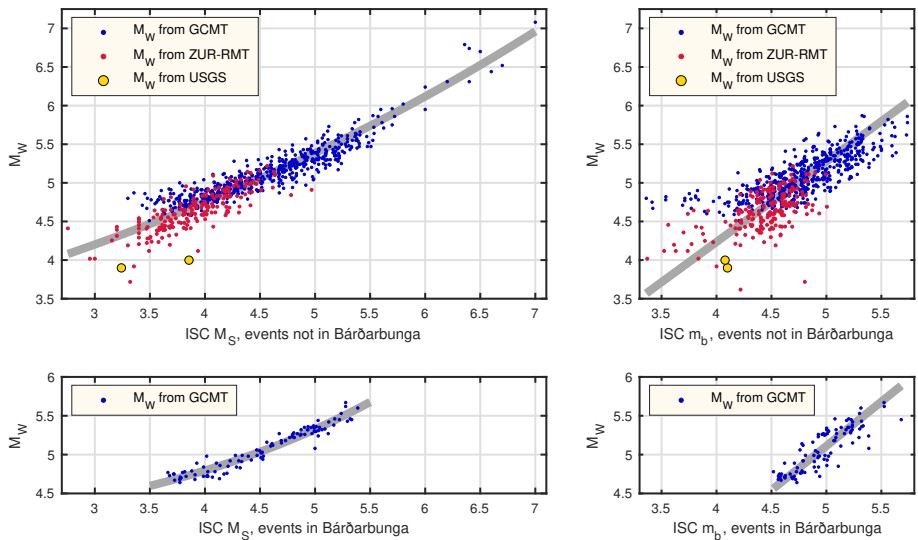

**Figure 3.** Magnitude pairs for earthquakes in the Northern Mid-Atlantic Ridge (NMAR) region 1976−2019, exponential relations for $M_S$ and linear relations for $m_b$, all fitted with $\chi^2$-regression. There are 733 $M_W$-$M_S$ pairs outside Bárðarbunga and 95 in it, and 744 $M_W$-$m_b$ pairs outside and 97 in Bárðarbunga. Note that a few earthquakes with $m_b < 3.5$, and thus not included in the final catalogue, are used for the regression. A slight random jitter has been applied to the pairs to improve the visual appearance of the graphs.

As one might expect the deviation in the $M_S$ model is considerably lower than in the $m_b$ model (Fig. 3). Thus $M_S$ is used to compute a proxy $M_W$ when it is available, for 4217 events in the NMAR region, of these 933 are in the ICEL region. In the
385   absence of an $M_S$ value the $m_b$ relation must be used, for 2954 events in NMAR, of these 379 are in ICEL. $M_S$ is available for almost all large earthquakes, the ones that are important for hazard assessment. Only three $m_b > 5$-values are used to compute proxy $M_W$ in the ICEL region and therefore the regression only uses data with $m_b < 5.5$ (Fig. 3).

To use a somewhat round number, and to have a single $M_W$ uncertainty, the current work uses $\sigma(M_W) = 0.09$ for all the models, $m_b$ and $M_S$, in and outside Bárðarbunga (Fig. 3, Table 1). These uncertainty values are in good agreement with the
390   results quoted in Sect. 4.2.1, perhaps somewhat lower, which might reflect that our data is more recent and there is continuous improvement in the quality of the global catalogues.

To study possible change in the $M_S$-$M_W$ relationship or in the accuracy of the moment tensor $M_W$ values, a separate modeling was tested for a few sub-periods. A slight, somewhat erratic, improvement in the accuracy was observed, but no significant change in the relationship. Thus it was decided to use a single model for the whole period.




**Table 1.** Parameters of exponential and linear models for $M_W$, obtained with $\sigma(M_W) = 0.09$, c. f. Eqs. (5) and (6), RMSD is the root-mean-square deviation between the model and the y-coordinates of the data, and the last column gives the estimated $\sigma(m_b)$ and $\sigma(M_S)$, respectively.

| Model | $a$ | $b$ | $c$ | RMSD | Uncertainty |
|---|---|---|---|---|---|
| non-caldera $M_W \sim M_S$ | 0.850 | 0.143 | 0.612 | 0.142 | 0.174 |
| non-caldera $M_W \sim m_b$ | 0.077 | 1.040 | | 0.256 | 0.225 |
| caldera $M_W \sim M_S$ | -0.961 | 0.322 | 3.410 | 0.070 | 0.008 |
| caldera $M_W \sim m_b$ | -0.602 | 1.143 | | 0.155 | 0.112 |

### 4.4 Uncertainty of the proxy magnitudes

Following Gasperini et al. (2013), the variance of $M_W^{\text{proxy}}$ for an earthquake obtained with $M_S$ regression may be estimated with:

$$\sigma_{\text{proxy}}^2 = (f'(M_S)\sigma_{\text{MS}})^2 + \sigma(M_W)^2$$
$$= \exp(a + bM_S)^2 \sigma_{\text{MS}}^2 + \sigma(M_W)^2 \tag{7}$$

where $\sigma_{\text{MS}}^2$ is the variance estimate for the earthquake, obtained as described in Sect. 4.2.2, $\sigma(M_W) = 0.09$ as in Sect. 4.3, $f$ is the model function given in 5, and $a$ and $b$ are the regression parameters (Table 1). The values of $\sigma_{\text{proxy}}$ computed with Eq. (7) are in the range $0.125 - 0.245$, and their RMS-average is 0.146, indicating that only few earthquakes have uncertainty in the high end of the range. A similar procedure is used in the $m_b$ regression case and the uncertainties given by the analog of Eq. (7) are in the range $0.256 - 0.527$ (RMS-avg. 0.288). For the caldera models, the uncertainty ranges are $0.102 - 0.177$ (RMS-average 0.113) for $M_S$ and $0.277 - 0.391$ (RMS-avg. 0.281) for $m_b$.

### 4.5 Uncertainty in recent local magnitudes

The SIL system described in Sect. 2.2.1 provides two types of local magnitudes, denoted with $M_L$ and $M_{LW}$. To assess the uncertainty in these values $\chi^2$-regression has been applied, with modeled (non-proxy) $M_W$-magnitudes on the y-axis and $M_L$ and $M_{LW}$ on the x-axis with $\sigma(M_W) = 0.09$, as in Sect. 4.3 (with caldera earthquakes excluded). The resulting estimates are $\sigma(M_L) = 0.471$ and $\sigma(M_{LW}) = 0.570$, far higher than the corresponding values 0.176 for $M_S$ and 0.225 for $m_b$. Restricting the comparison to earthquakes onshore Iceland (24 events) gave an improved $\sigma(M_L) = 0.224$ but a worse $\sigma(M_{LW}) = 0.748$. In all cases there is a considerable negative bias of $0.6 - 1.4$ magnitudes, more offshore (outside the SIL network) than onshore. One explanation for the large spread and bias of the local magnitudes is that the SIL systm's analysis is optimised towards robust magnitude estimation of smaller earthquakes than those of this comparison. Figure 4 shows the spread of the data, evidently in line with these estimates. It has no meaning to show the regression curves because of the high uncertainties.



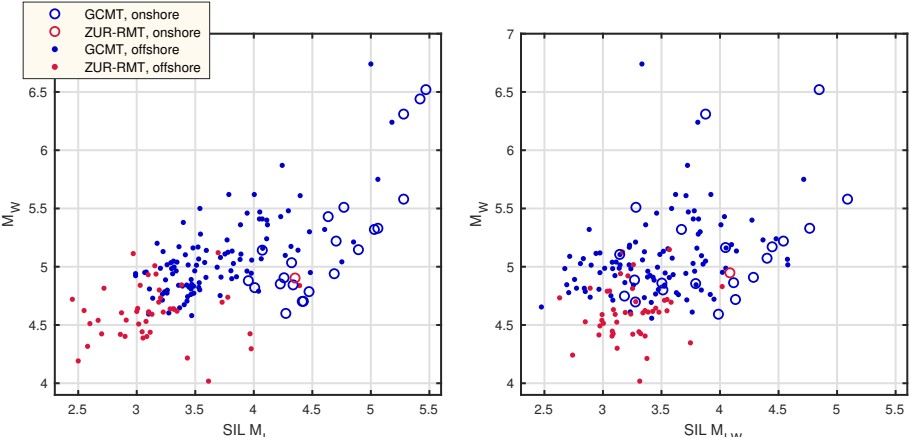

**Figure 4.** Moment tensor modeled magnitudes ($M_W$) and two types of local magnitudes computed by the SIL system (see Sect. 2.2.1). Earthquakes in the calderas Bárðarbunga and Katla have been excluded, but apart from that all events with both SIL- and $M_W$-magnitudes are included, 24 onshore and 146 offshore. The ZUR-RMT $M_W$ values were computed by the Swiss Seismological Service 2000–2005.

## 5   Results and discussion

The primary results of this study is the ICEL-NMAR catalogue, described briefly in the next subsection. Section 5.2 discusses the completeness of the catalogue as a function of magnitude and time. Next is a section which compares the new catalogue with the ISC-GEM catalogue discussed in the introduction, and finally there is a section with a general discussion. The catalogue earthquakes within the region $63°-67°$N and $13°-25°$W are plotted in Fig. 5.

### 5.1   The ICEL-NMAR Earthquake Catalogue

The new catalogue is available in the Mendeley Data Repository, as the *ICEL-NMAR Earthquake Catalogue*; see Data Availability section below. There are three files, `icel-nmar.txt` with the actual earthquake data, `supporting-info.txt` with meta information, and `sil-time.txt` with SIL origin times for comparison. For each earthquake `icel-nmar.txt` provides region (ICEL or NMAR), origin time, location, $M_W$, the $M_W$ uncertainty estimated with Eq. (4) or (7) as appropriate, and information on how the $M_W$ value is computed or what its source is. When available, similar information for $M_S$ and $m_b$ are given, and finally information on the origin time and location sources. All events smaller than $M_W$ 4 were excluded and the uncertainty was not computed for $M_W < 4.5$. The available information on hypocentral depth is very inconsistent and it is not provided in the catalogue. The brittle part of the Icelandic crust in most areas is less than 12 km thick, and earthquakes of any significance will rupture the whole thickness (Hjaltadóttir, 2010; Pedersen et al., 2003; Stefánsson et al., 1993).


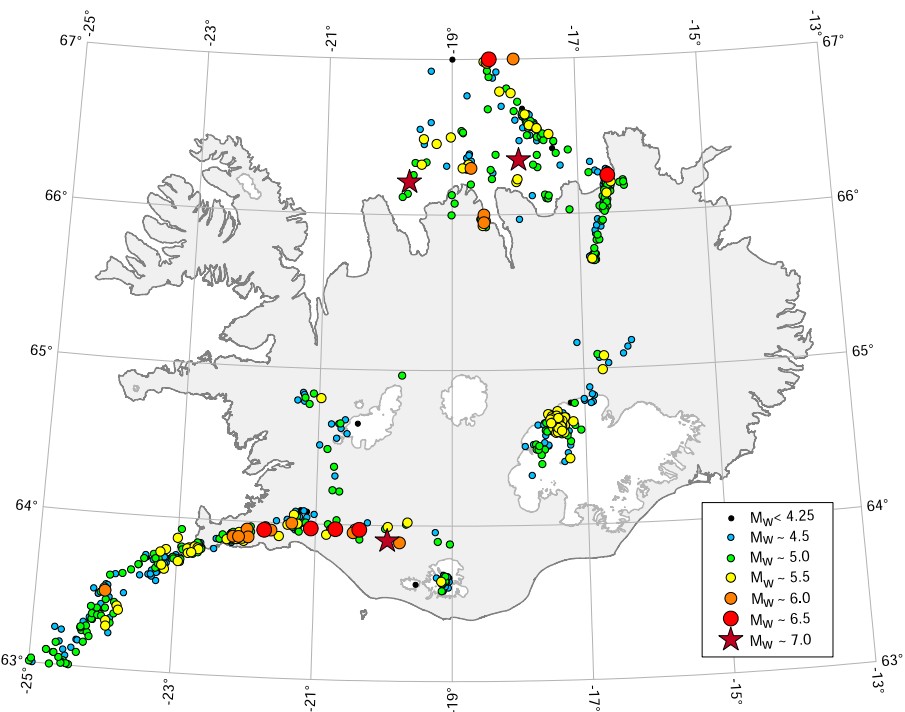

**Figure 5.** Earthquakes in or near Iceland during $1900-2019$ listed in the new catalogue. For the first part of the period, location coordinates are often given in round numbers (tenths of degrees or even half or whole degrees). The map shows slightly jittered locations ($\leq 3$ km; except when $M_W > 5.75$) to avoid superimposing different events. The magnitude range for the smallest earthquakes is $M_W$ $4-4.25$. For the other ranges the central value is specified, so that e. g. $M_W \sim 4.5$ implies the range $4.25-4.75$. The largest event is $M_W$ 7.01 in the TFZ at $18°$ W.

## 5.2 Magnitude of completeness

To investigate the magnitude of completeness of the new harmonised catalogue for the whole NMAR region, two methods were used. Firstly, histograms with $10-30$ year bins of the earthquake count with magnitudes exceeding different thesholds were created (Fig. 6), and secondly Gutenberg-Richter models were constructed for a few selected periods and minimum magnitudes. The histograms show that the catalogue appears to be complete for $M_W \geq 6$ for the whole period, for $M_W \geq 5.5$ since 1915, for $M_W \geq 5$ since 1970, and for $M_W \geq 4.5$ since 2000. Gutenberg-Richter modeling with simple declustering (Gardner and Knopoff, 1974) indicate a magnitude of completeness of 5.5 for the whole period, and 4.5 for the period after 1970 (data not shown). For the ICEL region similar histograms indicated a completeness magnitude of 5.5 for the whole period, 5 from 1915, and 4.5 from 1965.

It is interesting to compare the number of large events during the 20th century with lists of historical earthquakes in earlier centuries. Table 2 shows earthquakes with estimated magnitude $\geq 6$ in Iceland or within 20 km offshore during $1700-1899$, in total 17 events. In the new catalogue there are 8 earthquakes with $M_W \geq 6$ in the 20th century in the same region, and 4 more in the first two decades of the 21st century.



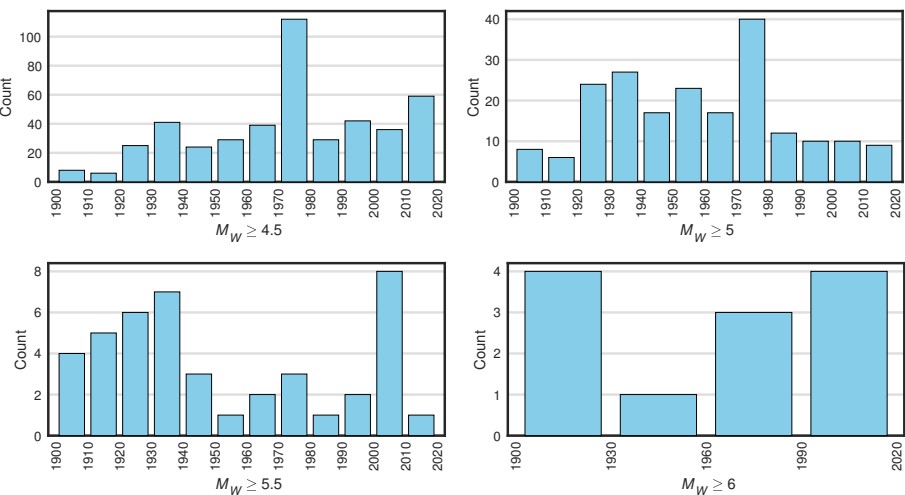

**Figure 6.** Count of earthquakes in the NMAR region exceeding different $M_W$ thresholds according to period.

In the final catalogue there are a few periods with disproportionately many earthquakes connected to tectonic activity (SISZ 2000 and 2008) and volcanic activity (Krafla region 1975–1976, Hengill 1994–1999, Bárðarbunga 2014–2015).

In the wake of large earthquakes it is possible that other events are triggered by their probagating waves. These secondary events can be missing from the international catalogues because their signal is lost in the coda of the primary event at teleseismic distances. An example of this are two events on the Reykjanes Peninsula triggered by the $M_W$ 6.52 South-Iceland event on 2000-06-17 15:40:41, occurring 26 and 30 seconds later, and 65 and 80 km farther west, respectively. The size of the first one was estimated to be $M_L$ 5.5 (Antonioli et al., 2006), and that of the second one $M_W$ 5.79 (Pagli et al., 2003). Our estimated $M_W$ for the first event is 5.5, and both $M_W$ values have been added to the new catalogue with uncertainties of 0.4 and 0.2, respectively. These are the only events not coming from one of the four international catalogues of Sect. 2.1.

### 5.3 Comparison with the ISC-GEM catalogue

Version 7.0 of the ISC-GEM catalogue was released in 2020. In the NMAR region it contains much fewer events than our new catalogue (168, with $M_W$ in the range 5.42–7.00), and no local information is used to relocate them. Non-proxy $M_W$ magnitudes in ISC-GEM and the current catalogue are identical, but in general the proxy values differ, both because ISC-GEM uses a different regression model and because the underlying $M_S$ and $m_b$ data may differ. The difference in the more important $M_S$ regression curves is slight. Comparing Fig. 3 and the corresponding figure in (Di Giacomo et al., 2015) for $M_S = 5$ the ISC-GEM curve is 0.06 higher, for $M_S = 6$ it is 0.02 lower and for $M_S = 7$ it is 0.05 lower.

There are 119 earthquakes with proxy $M_W$ common to the catalogues, of these 30 in the ICEL region. Their ISC-GEM magnitudes are on average 0.06 lower than the ones presented here. The largest absoloute difference is 0.47 and for 85 events the difference is less than 0.2. For the ICEL region the mean difference is 0.02, the largest absolute one is 0.26, and there are 24 events which differ by less than 0.2 magnitudes.




**Table 2.** Historical large earthquakes in Iceland in the 18th and 19th centuries. The magnitude estimates are based on the resulting damage (Halldórsson, 1992b; Stefánsson et al., 2008; Sólnes et al., 2013). The epicentral locations are approximate but overall the longitude is more accurate than the latitude since in most cases N-S surface faults have been mapped and linked to the largest events. Note that these earthquakes are not included in the new catalogue.

| Date | Lat. | Lon. | $M_S$ | $M_W^{\text{proxy}}$ |
|------|------|------|-------|----------------------|
| 1706, April | 63.9 | 21.2 | 6.0 | 6.1 |
| 1732, Sept. | 64.0 | 20.0 | 6.7 | 6.7 |
| 1734, March | 63.9 | 20.8 | 6.8 | 6.8 |
| 1755, Sept. | 66.1 | 17.6 | 7.0 | 7.0 |
| 1766, Sept. | 63.9 | 21.2 | 6.0 | 6.1 |
| 1784, August | 63.9 | 20.5 | 7.1 | 7.1 |
| 1784, August | 63.9 | 21.0 | 6.7 | 6.7 |
| 1829, Feb. | 63.9 | 20.0 | 6.0 | 6.1 |
| 1838, June | 66.3 | 18.8 | 6.5 | 6.5 |
| 1872, April | 66.1 | 17.4 | 6.5 | 6.5 |
| 1872, April | 66.2 | 17.9 | 6.5 | 6.5 |
| 1885, Jan. | 66.3 | 16.9 | 6.3 | 6.4 |
| 1896, August | 64.0 | 20.1 | 6.9 | 6.9 |
| 1896, August | 64.0 | 20.3 | 6.7 | 6.7 |
| 1896, Sept. | 63.9 | 21.0 | 6.0 | 6.1 |
| 1896, Sept. | 64.0 | 20.6 | 6.5 | 6.5 |
| 1896, Sept. | 63.9 | 21.2 | 6.0 | 6.1 |

A few events which differ most were investigated, and it transpired that the explanation was usually a combined effect of the regression curve difference and the underlying data difference.

## 5.4 Cumulative seismic moment and the earthquake cycle

The question arises how representative the seismic activity of the catalogue period is for any period of 120 years. The answer depends on the length of the typical earthquake cycle. If the cycle is significantly longer than 120 years our sample may underestimate the seismicity greatly, e. g., if the period does not contain a characteristic maximum magnitude earthquake. Studies of South Iceland earthquakes indicate that we may be near this critical duration of the cycle. The study of Einarsson et al. (1981) gave an average time between major earthquake sequences of about 80 years, ranging between 45 and 112 years. Stefánsson and Halldórsson (1988) concluded that the South Iceland Seismic Zone (SISZ) had a total release of accumulated strain in about 140 years. Decriem et al. (2010) estimated the accumulated strain by plate movements since the 1896−1912 earthquakes and compared to the released seismic moment during the earthquakes of 2000 and 2008. They found that only about half of the strain had been released by these events.


For comparison with our catalogue we estimate the potential seismic moment release in the two fracture zones, the SISZ and the TFZ, by a simplified geometric model of two transform faults parallel to the relative plate motion. The simplification is justified by the arguments of Sigmundsson et al. (1995), who showed that the seismic moment of many closely spaced, short transverse faults (bookshelf faults) is equivalent to that released by a single transform fault. We also assume that almost all the seismic moment is released by the transform zones and not by the divergent segments of the plate boundary or the magmatically

induced seismicity. The length of the transform zones is taken as 180 km and 150 km for the South and North Iceland zones, respectively, i. e. the offset of the ridge axes. The width of the fault is taken to be the thickness of the seismogenic part of the crust, about 10 km, the spreading rate is 19 mm/yr, and the shear modulus $20 \cdot 10^9$ Pa (McGarr and Barbour, 2018). The moment rate will then be:

$$20 \cdot 10^9 \times 19 \cdot 10^{-3} \times 330 \cdot 10^3 \times 10 \cdot 10^3 = 1.25 \cdot 10^{18} \, \text{Nm/yr}. \tag{8}$$

This result can be compared with the total seismic moment released in Iceland during the catalogue period, which may be estimated using the catalogue data and the completeness information of Sect. 5.2. Such computation for all earthquakes $\geq M_W$ 4 in the area shown in Fig. 6, excluding the Reykjanes Ridge and Bárðarbunga, gives a total of $1.61 \cdot 10^{20}$ Nm. Adding a simple correction for smaller events assuming the Gutenberg-Richter law with $b = 1$ raises the estimate to $1.64 \cdot 10^{20}$ Nm, corresponding to an annual rate of $1.37 \cdot 10^{18}$ Nm/yr. This agrees quite (even surprisingly) well with the result of Eq. (8).

**5.5   General discussion**

We have constructed a new catalogue of earthquakes in Iceland and, as a byproduct, for the Northern Mid-Atlantic Ridge. A general criteron for entry into the catalogue is that an earthquake has been instrumentally recorded by agencies outside Iceland. Locations of events in the ICEL region (Fig. 1) have been reassessed and proxy $M_W$ values for earthquakes without modeled moment magnitudes have been computed. The resulting moment magnitudes range from 4 to 7.08. For the ICEL region the

catalogue is reasonably complete for $M_W \geq 5.5$ for the whole period. There are 36 earthquakes of this size onshore or less than 20 km offshore, i. e. 2.8 per decade, and of these 10 have $M_W \geq 6$, i. e. 0.8 per decade.

    To our knowledge, the map in Fig 5 is the first earthquake map of Iceland which is not substantially confounded by misplaced events. The locations of the two large TFZ-events marked with a star in Fig. 5 (the easternmost 1910 and the westernmost 1963) are still uncertain and controversial. Neither of them appears to have occurred on the best known structures, the Húsavík-Flatey

fault or the Grímsey Oblique Rift. Stefánsson et al. (2008) suggest that the 1963 event originated on a NNE-striking fault offshore Skagafjörður, based on the distribution of recent earthquakes and the focal mechanism solutions of Stefánsson (1966) and Sykes (1967). They furthermore suggest that the 1910 event originated on the eastern margin of the Grímsey Shoal. We adopt these locations in our catalogue. Distribution of epicentres and recent bathymetric data support these suggestions (Einarsson et al., 2019).

The largest events occur in the two seismic zones, where the plate boundaries are parallell to the plate movements (Fig. 1 and 5). The distance from these events to the Reykjavik capital area, where 63% of the population live, is some tens of kilometers, and the same holds for Akureyri in North Iceland, with 5% of the population. However there are several towns and villages



within the zones. An important future task is to carry out a detailed analysis of the seismic hazard both in these urban areas and elsewhere in Iceland. The new catalogue should prove to be an essential resource for such seismic hazard mapping.

*Data availability.* The international earthquake catalogues from USGS, GCMT and ISC are freely available online. In addition we used the catalogue of Ambraseys and Sigbjörnsson (2000), as well as scattered data on individual earthquakes from various printed sources, as detailed in Sect. 2. We also used the Icelandic Meteoralogical Office catalogue for the period 1926–2019. Work is currently underways to put at least part of this catalogue online. The new catalogue is available in the Mendeley data repository as the ICEL-NMAR Earthquake Catalogue (http://dx.doi.org/10.17632/7zh6xg22cv.1).

*Author contributions.* K Jónasson and B Bessason prepared the manuscript with contributions from all authors, K Jónasson and Á Helgadóttir wrote software for data processing, P Einarsson, GB Gudmundsson, K Jónasson, B Bessason, B Brandsdóttir and Kristín Vogfjörd reappraised event locations.

*Acknowledgements.* We thank all the people and institutions who have set up seismometers, gathered data from these, and used them to compute earthquake locations and magnitudes. Without their contribution this work would not have been possible. The public datasets that
we have used are available from three open web sources: the ISC Bulletin Event Catalog (2020); the GCMT Earthquake Catalog (2020); and the USGS Earthquake Catalog (2020). Finally, we use the earthquake catalogue of the Icelandic Meteorological Office for the period 1926–2019.





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
