# Peer review of "A Harmonised Instrumental Earthquake Catalogue for Iceland and the Northern Mid-Atlantic Ridge"

_Natural Hazards and Earth System Sciences, 2021_

## Referee Comment (RC2)

[referee-annotated manuscript omitted]

---

## Author Comment (AC1)

**Please see our responses (in blue) to referee #1 between his comments below.**

**Formal manuscript rating and recommendation to the editor** (non-public)
**1) Scientific significance**
Does the manuscript represent a substantial contribution to the understanding of natural hazards and their consequences (new concepts, ideas, methods, or data)?
**Excellent**
Good
Fair
Poor

**2) Scientific quality**
Are the scientific and/or technical approaches and the applied methods valid? Are the results discussed in an appropriate and balanced way (clarity of concepts and discussion, consideration of related work, including appropriate references)?
**Excellent373**
Good
Fair
Poor

**3) Presentation quality**
Are the scientific data, results and conclusions presented in a clear, concise, and well-structured way (number and quality of figures/tables, appropriate use of technical and English language, simplicity of the language)?
Excellent
**Good**
Fair
Poor

For final publication, the manuscript should be
accepted as is.
accepted subject to technical corrections.
**accepted subject to minor revisions.**
reconsidered after major revisions:
    I am willing to review the revised paper.
    I am not willing to review the revised paper.
rejected.

**Subject: Comment on nhess-2021-15**
Catalogs that are compilations from previously published catalogs, assembled according the modern criteria, are important for both tectonic studies and seismic hazard studies. The catalog resulting from this paper will play in important role in future studies focussing on Iceland and vicinity. The paper comes across as sound from a scientific perspective, although I am asking for more detail on several points to confirm this impression. I also see the desirability for revision to address issues of style and exposition.

> We thank the referee for his positive feedback. We shall do our best to address all the issues raised in his comments.

For discussion of prior cataloging of Iceland earthquakes, I would recommend explicit mention of the International Seismological Summary (ISS), the predecessor of the ISC. As is apparent from the results of the online search of the ISC (http://www.isc.ac.uk/iscbulletin/search/bulletin/ ), many of the early "ISC" locations are actually ISS locations. The ISS volumes have been scanned and put on-line by Italy's INGV (http://storing.ingv.it/ISS/index.html). Some of the ISS origins are actually those computed by IMO, but the ISS has, with these origins, associated arrival-times (actually, travel-times computed with respect to the published origin-times) from stations world-wide.

> We shall make a mention of the ISS in the revised manuscript: **"Another global source for earthquakes in the first part of the 20th century is the International Seismological Summary (ISS), the predecessor of the**

**ISC."** [line 58]. Our web search indicates that there are no magnitudes in the ISS bulletins. There are few earthquakes from the ISS period that we take from the ISC catalog, and most of those are marked "Gutenberg-Richter".

Following are comments on particular sections or lines of the paper, referenced to the line number on the PDF file that was provided for review. I would acknowledge that some of these comments do not identify issues errors or issues that are important for understanding the paper, but just my personal stylistic preferences. I am assuming that the authors will recognize these cases of personal stylistic preference and judge for themselves whether or not to address them,

line 6 — "modified by some expert judgement" — this phrase should be revised to provide a better sense of the "expert judgement". Presumably, in the context of the sentence in which it resides, the expert judgement is based on something besides technical reports, scientific publications, and newspaper articles. A possible example of modification by expert judgement, which should be possible to explain, would be the reinterpretation of some previously ambiguous data in terms of seismological or tectonic understanding that has been acquired since the data were initially interpreted

To clarify and simplify the description of the methodology used, we propose/intend to remove the reference to "expert judgement" from the abstract (viz: "...scientific publications, **and** newspaper articles, and modified by some expert judgement. The catalogue contains..."). The expert judgement applies to events for which 6 of the authors (cf. section "Author contributions") reappraised event locations at meetings. Sometimes the sources are explicitly vague about the location, and sometimes the sources' epicenter is evidently wrong. Thus the example phrase given by the refree might adequately apply to both cases, and we intend to add a sentence at the end of the first paragraph of section 3, where "expert judgement" occurs again: "**One could say that we have reinterpreted the data with seismological and tectonic understanding that has been accumulating in recent years and decades**."

line 8 — The authors citation of the largest magnitude of Mw 7.01 will seem naively precise to many readers, for reasons that are discussed at length in the body of the paper. I would recommend giving the "largest magnitude" to lower precision in this instance.

Indeed. We shall make the change.

lines 8-9 — I would suggest revising the description of use of local and teleseismic data, to account for the fact that listings in the ISC and ISS have made use of both local and tele seismic data.

We shall change "local and teleseismic data" to "**local data and teleseismic catalogues**" on line 8 in response to this suggestion. Just to clarify, we state earlier in the abstract that we are combining local epicenter information and global magnitude information, and the phrase "melting" refers to this. Few of the ISC magnitudes are based on local data, and we explicitly disregard Reykjavik and Akureyri as source agencies in our scripts.

line 9 — I suggest leaving out or modifying the second clause. Previous catalogs, such as those of the ISC and ISS, would not have contained epicenters that were obviously mislocated in ways different from the ways in which the epicenters of the ICEL-NMAR catalog are mislocated.

We are not suggesting that the nature of mislocation is different, but instead its magnitude. The accuracy of the locations in the new catalog is much better than in earlier maps, which sometimes (or often) provide locations that are wrong by tens of kilometers, and place earthquakes in locations where it is absolutely certain that they did not occur (such as directly under Reykjavík).

We intent to change "with no obviously mislocated events" to "**with much more accurate locations than earlier maps**" to try to be clearer.

line 11 — "computed with chi-squared — regression". The proxy Mw values themselves are not computed with chi-squared regression, as is literally stated. The proxy Mw values are calculated from equations that were determined with chi-squared regression.

OK. We intend to change "with" to "**using**"

line 12 — "All the presented magnitudes have associated uncertainty estimates" I suggest revising, and augmenting, this sentence by citing typical, or example, values of uncertainty that are associated with a few classes of Mw.

> We agree, this is a good idea if the length limits on the abstract would allow. The MW-uncertainty is not estimated when MWproxy < 4.5, it is about 0.10 for moment-tensor modelled values, 0.15–0.20 when MW is computed from MS, and 0.25–0.35 when it is computed from mb. Accurate information on the estimates is already provided in section 4.4. Note that this means that the statement ("All the presented...") is incorrect and we shall correct it to: "**Magnitudes M ≥ 4.5 have associated uncertainty estimates**".

lines 12-14 — The conclusion on the relationship of seismic moment to plate displacement should be summarized more precisely and informatively, so that the reader can better anticipate the reasoning you have use in the text to relate seismic moment to plate displacement. For example, as discussed in the text, there is relatively little seismic moment associated with the rift (i.e., non-transform fault) section of the plate-boundary.

> To expand on the discussion of total sesmic moment and plate displacement in the abstract we shall add the clause "**indicating that the seismic activity of the catalogue period might be typical for any 120 year timespan**" to the second last sentence in the abstract, drawing on the discussion at the beginning of section 5.4.

line 17 — I think that what the authors call the "North-America plate" is usually called the "North American plate" and their "Euro-Asia plate" is usually called the "Eurasian plate". This was my impression, and I see it confirmed by the results of web-searches on the various alternatives.

> Good point! Thanks for the web-search. We shall make the change.

line 26 — typo — "where recorded" should be "were recorded".

> ok

lines 29-30 — I would suggest specifically mentioning the Reykjavik Mainka seismograph(s) at this stage of the manuscript, and providing a citation. (somehow, my word processor occasionally changes "Mainka" to "Maniac". If, in the version of this review that is transmitted to editor and authors, you see a reference to a "Maniac seismograph", please read that as "Mainka seismograph".) A suitable reference, although the year in which continuous operation resumed is given as 1926 instead of 1925 , might be Charlier and van Gils (1953), which can be downloaded at http://ds.iris.edu/seismo-archives/info/stations/Charlier1953.pdf .

> Much appreciated "research" by the referee, we shall follow his advice, and mention Mainka with a reference at the end of the first paragraph of the introdcution: "**In 1909 a Mainka seismograph was installed in Reykjavik. It was operated until 1914, and again from 1925 when continuous operation was secured (IMO: Vedrattan (the Weather), 1924–2006)**".

lines 31-37 — I would recommend, somewhere in this paragraph, stating that focal-depths are not given in the catalog, and explaining the reasons for this decision. This is currently discussed in lines 427-429, in which position it might come across as an afterthought.

> ok. We shall end the paragraph with: "**Icelandic earthquakes are almost always less than 12 km deep, but the exact depth information is often not resolvable and therefore the catalogue does not include hypocentral depth.**"

lines 39-41 — You later (lines 273 - 279) discuss Mw (ZUR-RMT). I am thinking that Mw(ZUR-RMT) should also be mentioned at this point (around lines 39-41).

> Here we are simply stating from which sources we obtain the data, not the original sources. We take all the ZUR-RMT that we use from the ISC online catalogue. Thus we do not intend to make changes to the manuscript in response to this comment.

lines 47-48 — This sentence does not do a satisfactory job of conveying why locations are the opposite of magnitudes. Most of the magnitudes in the catalog [particularly the Mw(GCMT)] are also based on teleseismic data, in contrast to the implication of the sentence. Moreover, errors in magnitude do not have the dimensions of distance that characterize location errors. So by what standard does one conclude that magnitudes are more accurate than

locations? I think the bottom line is that most of the uses for which the authors envision their catalog are more robust with respect to the likely catalog errors in Mw than to the likely catalog errors in location. But there are situations in which errors in magnitude can have more important consequences than errors in location. An example (although not pertaining to Iceland) would be the monitoring of nuclear-threshold treaties, in which anomalously high mb for a natural earthquake occurring within or near a national nuclear-test site may lead to suspicions that the nation that uses the test site is violating its signing of a treaty. In the case of the present ICEL-NMAR paper, I would recommend omitting discussion of relative accuracy of magnitudes and epicenters, and just focus on the reasons for using the Mw scale to express magnitudes of all earthquakes and on reasons for using local data to relocate the epicenters.

> We think that the reason for this referee comment is that our senctence is not clear. What we are trying to convey is that locally determined epicenters are more accurate than teleseismically determined epicenters, whereas globally determined magnitudes are more accurate than locally determined ones.
>
> We intend to ammend the two paragraphs before the one being discussed: (a) State on line 36 that the magnitudes are MW (...reappraised **MW** magnitudes...) and (b) begin line 38 with: **"The most accurate magnitude information comes from international catalogs. Therefore the magnitudes are all copied..."**.
>
> We also intend to rewrite the paragraph on lines 47–52: "**For the whole catalogue period local information is crucial for improving earthquake locations. Before 1955, and also for several subsequent events, written sources often provide valuable location information. Since 1955, when three seismometers were installed in Iceland covering the primary seismic zones, locally computed epicenters may be assumed to be more accurate than teleseismic epicenters in international catalogues, which are off by tens of km. One of the innovations in the new catalogue is therefore to use such local data. The primary local sources on epicenters are catalogues compiled at the Icelandic Meteorological Office (IMO), seismological bulletins, newsletters and reports published by the IMO and the University of Iceland Science Institute (UISI), journal articles with results of studies on Icelandic earthquakes, and contemporary accounts of earthquakes from newspapers."** We hope this will be sufficient to clarify.

line 48 — There is an implication that the use of local arrival-times automatically leads to hypocenters that are superior to those based on teleseismic data. I would note that many of the earthquakes relocated in this study are at a distance from Reykjavik such that their first arrivals at REY will be Pn waves, whose computed arrival times (used in the location process to interpret the observed arrival times) will be sensitive to such characteristics of the model as assumed crustal-thickness, upper-mantle P velocity, and degree of anisotropy. For some of the early earthquakes, REY is the only station that does not lie to the east of the epicenters, and bias in theoretical Pn arrival-times at REY could lead to substantial bias in the epicenters. So the fact that a hypocenter is based on local arrival-times does not automatically make it superior to those based only on teleseismic data. That stated, I would agree that data from REY are an important supplement to teleseismic data, and that epicenters assigned by REY (IMO), determined with REY data, or somehow constrained by REY data, are prime candidates for preferred epicenters.

> We hope to have addressed the concerns of the referee adequately by the proposed changes implied by our reply to the previous comment. We should emphasize that the article never *computes* locations from seismic measurements, but instead copies locations from other sources, sometimes complementing with judgement. This is is made very clear in section 3.1. Also, before 1955, there are only 9 out of 109 events with the Icelandic Meteorological Office specified as "location source" in the catalog and we find it unlikely that the epicentres have been computed from REY together with only more eastely teleseismic stations.

lines 51-52 —Similar to my criticism of "expert judgement" in line 6, I think it is generally unnecessary to make a general statement that the authors have used their judgement in interpreting previously published data. However, for cases in which the authors' judgements lead to seismotectonic inferences that are different than seismotectonic inferences previously made with the same or similar data, then the authors should indeed articulate, and take responsibility for, the specific judgements they have made in these cases.

> We shall remove the sentence refereed to, namely: "*These sources are complemented by the author's judgement*". We also note that the catalogue itself clearly indicates locations where authors have relocated events (marked "New" under location source).

Figure 2 — The caption should give information on the catalogue(s) that are the source of the plotted epicenters and magnitudes (i.e., are these some set, which should be described, of previously accepted epicenters and magnitudes,

or are they the epicenters and magnitudes resulting from this study?). Comparing this figure with Figure 5, it appears that the epicenters in Figure 2 are consistent with those of Figure 5, but that the magnitudes are not.

> The referee is absolutely correct, the caption is missing this info.. The locations and magnitudes are indeed those resulting from the current study. We shall add "**The displayed locations and magnitudes are those of the new catalogue.**" at the end of the caption. We shall also make sure that the magnitudes will be consistent between Fig. 2 and Fig. 5, by providing a new version of Fig. 2.

lines 74-75 — The assertion that the Grunthal and Wahlstrom (2003) magnitudes are systematically biased should be documented by citing a reference for this bias.

> We shall rephrase the paragraph, emphasizing that we are comparing Grunthal and Wahltrom's magnitudes with those of our manuscript. As we point out in the ms their data comes originally from the IMO (with a URL of the source given in the ms), and not from ISC or other international catalogs. Therefore there is no reference that can be cited apart from those already provided. We shall however add some details on the magnitude differences for three period bins: **the average difference is 0.41 before 1970, 0.37 between 1970 and 1980, and 0.27 after 1980; 3rd quartiles 0.59, 0.47 and 0.36 respectively**.

lines 80-81 — This sentence should convey the reason why the paper of Woessner et al. (2015) is evidence that previously discussed hazard maps for Iceland overestimate the hazard. Does the paper document a consensus among hazard mappers that the the earlier estimates were greatly overestimated for Iceland, or is it simply that inspection of the hazard map of Woessner et al. shows lower hazard than shown in the earlier maps. Also, discussion of bias in the hazard maps that used the catalog of Grunthal and Wahlstrom (2003) should convey whether the bias is due entirely to magnitudes being biased, or is some of the bias is due to some other assumptions used in preparation of the earlier hazard maps, such as assumptions concerning site response.

> We are sorry that we were not very clear. The Woessner paper is reporting on the SHARE results, with an estimated PGA for a 10% excedance probability in 50 years as 0.4–0.5 g in Reykjavik, and we deem this as being a big overestimate, and support this by referring to four publications which estimate a corresponding PGA of 0.1–0.2 g. Yes, some of the bias is probably due to other assumptions. We shall rephrase the whole paragraph to read:
>
> > **"For the Iceland region, all these projects adopted the original 2003 catalogue, adding data (locations and local magnitudes) after 1990 from IMO's catalogue. Among the products of these studies was the "SHARE" hazard map for Europe, where the hazard was greatly overestimated in some places in Iceland, among them in the Reykjavík capital area, where the estimated PGA for a 10% excedance probability in 50 years is given as 0.4–0.5 g (Woessner et al., 2015). Several recent local studies estimate 10% 50 year PGA as 0.1–0.2 g in the Reykjavik area (Sólnes et al., 2004; SCI, 2010; Sólnes et al., 2013; D'Amico et al., 2016). The reason for the presumed overestimation is likely a combination of errors in the underlying catalogs and differences in modelling."**

Section 2.1 — The catalog of Ambraseys and Sigbjornsson would not be an international catalog, if it only covers the region of Figure 1. It should discussed in the next session.

> We propose to change the title of section 2.1 to read "**Teleseismic catalogs**", in order to describe our catalog classification better. We shall also ammend the text in one or two other places to reflect this change.

Section 2.1 The Mw (ZUR-RMT) catalog should be mentioned in this section.

> We actually consider the ZUR-RMT tensors not to be a separate catalog, as they come from the ISC catalog, but we have added a mention of them under ISC in section 2.1.1: "**Among other important agencies is the Swiss Seismological Service, providing the ZUR-RMT (Zurich Moment Tensors)**". Instead we do not need to explain the abbreviation in section 4.1.1.

line 122 — The ISC did not exist before 1950. It's predecessor, the ISS, did exist. See http://www.isc.ac.uk/about/.

> Yes, the statement is a little inaccurate. We shall change it to emphasize that it were the earthquakes that happened before 1950, but the reporting of them happened later: "**epicentres of events before 1950 reported by the ISC**". Now it reflects better what the cited book says.

line 126 — Contrary to what is implied by this sentence, the USGS usually computes several magnitude types per earthquake, and these are given in some of the the output -formats of the USGS earthquake catalog-search that the authors cite. Also, the ISC on-line catalogs commonly attribute multiple magnitude types to the NEIC, which corresponds to the USGS. However, some output formats of the USGS/NEIC catalog-search do provide only one type of magnitude per earthquake. Also, for purposes of communicating with the media and the public, the USGS/NEIC does select a single magnitude value, so that the media do not get stirred up by the apparent "inconsistencies" of USGS magnitudes. Finally, there was a time when the predecessor to the USGS (NOAA) computed only mb values.

Yes, we agree, our description was confusing. We shall rephrase with the following paragraph: "**A simple online search in the USGS catalogue (2020) provides one magnitude value per earthquake (MW, MS or mb), although several magnitude types are often computed. The remaining values are in the ISC database, labelled USGS. Corresponding magnitudes from the two sources are in almost all cases identical. However the locations in the USGS catalog are different from those in the ISC catalog, the difference frequently amounting to a few tens of kilometers.**"

lines 132 — There is no mention of a Mainka seismograph in the current version of the Introduction. I have suggested (above discussion of lines 29-30) specifically mentioning the instrumentation in the Introduction.

Will be done!

line 225 — change "upto" to "up to" .

OK

line 254 — "and therefore the waveforms fit better" — this explanation for the reliability of teleseismic magnitudes would not apply to most magnitudes computed during most of the period covered by the catalog, computed from amplitudes and periods, but not based on waveform modeling.
We are not referring to source function modelling but simply the normal computing of MS and mb from the seismograms. To reduce the chance of misunderstanding we now say: "**The dominant periods at teleseismic distances are longer and the structure is smoother due to attenuation of the higher frequencies**"

line 257 — Similar to my comment on p. 11. It appears to me that, consistent with most studies that use the equivalent of a proxy Mw, the regression-determined equations that relate mb and Ms to Mw are determined from relatively recent earthquakes for which both the other magnitudes and Mw are independently available, and then these relations are used to determine the proxy Mw of the earlier events from the events' mb or Ms. The authors description of their methodology implies that the mb and Ms of the earlier earthquakes are somehow included in the process by which the regression-determined equations are obtained.

We think we understand the reason for this comment: We say on line 258, that we use a "larger collection of earthquakes" for the modelling. We did not imply the use of mb/MS from earlier earthquakes, but instead from a larger region. To clarify we have changed: "*a larger collection of earthquakes than is really needed in the Iceland context is used to construct...*" to "**earthquakes from the whole NMAR region are used to construct...**"

line 275-279 How are the ZUR-RMT determined? The use of "RMT" to describe these moment tensors suggests to me that they are determined with regional, rather than global, data. The methodology and the data used for the ZUR-RMT should be briefly summarized in the paper, with a reference provided to the source of the ZUR-RMT.

We shall add a reference to "Braunmiller et al. 2002" which seems to be the "original" ZUR-RMT article at the end of section 2.1.1 where we also intend to mention it and the Swiss Seismological Service (see above comment on Section 2.1). The R in RMT does indeed seem to indicate *regional*. We obtain these data from ISC, and thus it is may be misleading to refer to them as a catalogue on line 275; we shall therefore change "*are listed in both the GCMT and the ZUR-RMT catalogues*" to "**have both a GCMT value and a ZUR-RMT value".**

Figure 3 — caption, "improve visual appearance of the graphs" — I would recommend revising this reason to be more like that of the caption of Figure 5, which conveys the purpose of the jitter is to avoid superimposing different events (data points).

OK, shall be done.

line 384  —I would recommend changing "…events in the NMAR region, of these 933 are in the ICEL…" to "…events in the NMAR region, of which 933 are in the ICEL…" or "…events in the NMAR region:  933 of these are in the ICEL…".  This is a stylistic quibble.  The Reader (as did I) will know what you are trying to say.

OK, fine.

line 385 — Similar recommendation as that immediately preceding,  for the current phrase "…2954 events in NMAR, of these 379 are in ICEL."

OK, shall do.

line 390 — I would recommend changing "and there" to "and that there".

OK

Figure 5 — It would be desirable to plot in this figure the tectonic features and some of the geographic that are shown in Figure 1, to make it easier for the reader to assess the spatial relationship between the epicenters of the new catalog and the tectonic/geographic features.

We shall add both the boundaries of the seismic zones (the TFZ and the SISZ), as well as some of the main tectonic features from Fig 1 (the ridge and fault lines).

line 400 — I would suggest changing "5" to "equation (5)"

We change it to Eq. (5) which is the journal standard style

lines 427-429 —"available information on hypocentral depth is very inconsistent" —the meaning of this phrase is not clear.  The available information on hypocentral depths are consistent in implying that Icelandic earthquakes occur in the uppermost tens of kilometers of the earth's crust, as is implied by the next sentence of the paper.  What the available information cannot do is to generally resolve the depth distribution within the uppermost crust.  I do think that it is very important to state, as these lines do, that the catalog does not give estimates of focal-depth and that it is important to provide an explanation for not listing the focal depth.  See also discussion of lines 31-37.

We shall remove the unclear phrase. Note that we shall also add a little discussion on line 37 (see our reply to lines 31–37) .

lines 478-480 — It would be desirable to explicitly state that you are assuming that relative plate motion at depths above 10 km is accommodated entirely by seismogenic slip (rather than aseismic slip), in addition to stating the assumption that most of the slip is occurring in the transform sections of the plate boundary.

We shall indeed add a small explanation after "10 km": "**below which the slip is assumed to be aseismic**"

lines 486-504 — These conclusions demand the labeling of tectonic and geographic features in Figure 5, as recommended earlier.

We shall indeed improve Figure

line 505 — "parallel" is mis-spelled.

OK

References — are the cited IMO publications now scanned and available on-line?  If so, the on-line address from which they might be downloaded should be given in the citations to the publications

"The Weather" (Veðráttan) is online and shall add the URL of that to the references. Unfortunately the other two IMO publications are not (yet) on line.

line 623 —The Reference of Stefansson et al (1993) should include the title of the paper.

OK, well spotted

Comments on the "supporting-info.txt" file:

The citation (14) to the present paper in the "supporting-info.txt" file reflects an out-of-date plan for the publication of the paper.

> Shall be corrected.

I would suggest introducing a paragraph entitled something like "ON THE FOCAL-DEPTHS OF EARTHQUAKES IN THE ICEL-NMAR CATALOG," in which you state that estimates of focal-depth are not given in the catalog, conveying that available evidence points to these events occurring at depths shallower than ** km (whatever is your best judgement), and providing the reason for the omission of focal-depth estimates.

> --> Páll

In the DESCRIPTION OF CATALOG ENTRIES, MS should be defined as "Surface-wave magnitude" instead of "Surface magnitude".

> OK. We shall also do a similar correction in two places in the main manuscript

In the DESCRIPTION OF CATALOG ENTRIES, mb should be defined as "body-wave magnitude" instead of "body magnitude".

> Shall do. This time the main manuscript is ok

In the DESCRIPTION OF CATALOG ENTRIES, "loc-src", "epicenter location" should be shortened to simply "epicenter", since the additional descriptor "location" is redundant with respect to "epicenter".

> OK.

In the MAGNITUDE AND TIME SOURCES, it is not clear to me what is represented by "ISCother". For at least some of the events for which ISCother is listed as the source of a type of magnitude, there are many estimates of that magnitude-type listed in the on-line ISC catalog. Is the ISCother magnitude a mean or median of the different estimates of the magnitude-type?

> It is a technical difference explained on the ISC home page. We shall include a reference to that page in the supporting-info file and also rephrase the line in question and say:

```
ISC        The ISC online catalog [9], reviewed magnitudes
ISCother   The ISC online catalog [9], not reviewed magnitudes
           See http://www.isc.ac.uk/iscbulletin/search/bulletin/
```

The Internet address of USGS online catalog should be given somewhere in "supporting-info.txt".

> We shall add that, and also the URL for the GCMT catalogue.

---

## Author Comment (AC2)

This submission describes the work to compile an update instrumental earthquake catalogue (since 1904) for Iceland (ICEL) and surrounding areas, including a large part of the North Mid-Oceanic Atlantic ocean (NMAR region). The authors combine several sources, both from local and international providers. For what regards location parameters, the authors main focus has been to find the best source for different time periods, and validating cases using, when available, local information and their judgement, given the knowledge of the area. Note that for the early part of last century this approach can introduce in the catalogue mislocated events (many entries in the early years are rounded to the nearest degree) and, at times, spurious entries, as pointed out, for example, by Di Giacomo and Dewey (2020).

Most of the focus of the work, however, is to harmonize the magnitude composition of the catalogue in terms of Mw. When available from GCMT, the authors use it as golden standard, otherwise a proxy value is computed. The authors, therefore, derive conversion relationships for the area using MS and mb, preferably from ISC, or a mix of available values. Whilst this approach is sound, the uncertainties of the proxy values found in the catalogue seem somewhat optimistic, particularly for those entries where the authors use MS from different sources.

The work in general is well presented and the text requires only minor adjustments. However, (possibly) some inconsistencies in the catalogue files have been found and some questions on the authors approach are raised. Please see the comments, concerns and queries in the annotated PDF here attached.

We thank the referee for his kind words, and even more for his very careful reading of the manuscript and meticulous checking of the accompanying data files.

Regarding the uncertainties provided for the proxy values, they are simply statistical estimates, computed with the statistical methods described in the manuscript, using as input uncertainty estimates from the cited literature as well as properties of the data themselves. Thus they are aleatoric and no attempt is made to guess or estimate the epistemic part of the uncertainty (cf. the Wikipedia article on "Uncertainty quantification"). Nevertheless we can admit that the uncertainties may be somewhat optimistic, especially for the mb regression. We shall thus change 0.30 to 0.35 on line 332 (mb error before 1965).

We have copied the annotations from the PDF file, and they are listed below with our replies after each one.

L7:  First occurrence of Mw, add its full name here and use Mw in place of "moment magnitude" at line 11
     OK

L8:  I suggest to use 7.0 instead of 7.01 in the text
     Shall be done

L8:  Consider replacing "melting" with "merging"
     Yes, we replace it

L12: the surface wave magnitude MS, but exceptionally on the short-period body-wave magnitude mb
     We shall put in surface-wave and body-wave but exclude short-period, to not lengthen the abstract. We also put the definition (i.e. MS, mb in brackets) in the Introduction and not here for the same reason.

L31: With the the first occurrence of "harmonized earthquake catalogue" in the text, it would be appropriate to define what "harmonized" means here. Recent literature on this topic often uses "harmonized" as "homogenized". Hence the reader may expect that the authors harmonize (homogeneise) the catalogue both in location and magnitude. However, the authors harmonize only in terms of magnitude, and the locations are compiled from several different source
     We shall add the suggested definition of harmonised, and also emphasize that it are only the

magnitudes which are harmanized. We plan to state: "**...construct a catalogue with harmonised magnitudes (which are comparable in both time and space) and reassessed locations**"

L39:   Please use the citations given at https://www.globalcmt.org/CMTcite.html
We do provide those citations further down where the GCMT is discussed under its own heading. Here there is only a reference to the GCMT website. Similarly, proper citations to ISC wait for the section about ISC. We are trying to make the Introduction more readable. Hope that is OK.

L47:   Delete "Opposite to magnitudes"
This paragraph has been (completely) rewritten in response to critique by Referee 1 (including dropping "Opposite to magnitudes"), see our reply marked "lines 47–48" above.

L48:   Is this intended for the entire catalogue? It would help if the authors specify in which time period "local information" is crucial to improve the locations
The rewritten paragraph now clearly states that it is intended for the entire catalogue (see rewritten paragraph marked "lines 47–48" above)

Fig 2:   [*Pencil around largish earthquake south of Ireland*] I suggest to filter out from the NMAR region all earthquakes that are not part of the mid-oceanic ridge. Indeed, there is no need to include in this catalogue earthquakes that occurred in UK and France, and also the event along lat=45N and east of longitude -20. In addition, did the author check the reliability of event 0674 in the catalogue file (orange circle south the coast of Ireland, highlighted in the map by a hand-drwan green marker)? It seems that the authors picked from the ISC Bulletin a NAO solution (hence not clear why loc-src and time-src in the catalogue file are set equal to ISC, they should be ISCother), which is unreviewed and probably should have been grouped with the explosion at 08:33:46, see http://isc-mirror.iris.washington.edu/cgi-bin/web-db-v4?request=COMPREHENSIVE&out_format=ISF&searchshape=RECT&bot_lat=&top_lat=&left_lon=&right_lon=&ctr_lat=&ctr_lon=&radius=&max_dist_units=deg&srn=&grn=&start_year=1971&start_month=7&start_day=20&start_time=08%3A00%3A00&end_year=1971&end_month=7&end_day=20&end_time=09%3A00%3A00&min_dep=&max_dep=&min_mag=&max_mag=&req_mag_type=&req_mag_agcy=&min_def=&max_def=&include_magnitudes=on&include_links=on&include_headers=on&include_comments=on

All events outside the ICEL region are taken verbatim from the ISC catalog, and no reliability check is made (as we state in the manuscript). This includes event 0674. The ISC catalog (downloaded from the ISC website cited on line 97 on 23 Nov. 2020), contains codes for each magnitude on whether it is reviewed or not, but no such information is provided for the origin time and epicenter. This explains why sometimes a magnitude source is given as "ISCother" and sometimes as "ISC", but all times and epicenters from ISC are simply coded as ISC. Section 2.1.1 in the manuscript explains how the ISC catalogue is organized. We shall reorganized and improved the explanations in the supporting info file and say:

> **For each event the ISC catalog [12] has one location and time but multiple magnitudes. Some of the magnitudes are marked ISC, and these have been reviewed by the ISC. Magnitudes marked CTBTO, PAS, Sykes, ZUR-RMT and most of the USGS-ones also come from the ISC-catlog, and so do those marked ISCother, which covers all remaining non-reviewd ISC magnitude sources.**

> **Locations and times taken from the ISC catalogue are, however, all marked ISC.**

Regarding the suggestion to filter out events outside the NMAR, we are hesitant. Firstly, it would have a very marginal effect on our regression models to exclude these events, as they are few and none of them big. In addition, the map in Figure 2, and the corresponding data in our catalogue, simply state what is in the ISC catalogue, without any judgement, and we think it may be informative to some readers.

L68: The ISC Bulletin contains the local solutions (under agency REY), and local station readings since its beginning in 1964. As such, the ISC Bulletin combined local and global solutions, see events from http://isc-mirror.iris.washington.edu/cgi-bin/web-db-v4?request=COMPREHENSIVE&out_format=ISF&searchshape=RECT&bot_lat=62&top_lat=68&left_lon=-26&right_lon=-12&ctr_lat=&ctr_lon=&radius=&max_dist_units=deg&srn=&grn=&start_year=1964&start_month=1&start_day=01&start_time=00%3A00%3A00&end_year=1999&end_month=1&end_day=01&end_time=00%3A00%3A00&min_dep=&max_dep=&min_mag=3.8&max_mag=&req_mag_type=&req_mag_agcy=&min_def=&max_def=&include_magnitudes=on&include_links=on&include_headers=on

That is correct (but there are only 14 events given a REY location, in 2018–2019). However after staring at our statement we decided to remove it (it does for example not specify Iceland).

L75: magnitude units
OK "**magnitude units**"

L82: 1904
Absolutely right

L82: replace with "Global Instrumental Earthquake Catalogue"
OK. We shall spell everything out in full.

L83: Note that version 7 of the ISC-GEM catalogue was released in April 2020. It may be that the authors used version 6 (released in 2019) at the time of writing this work / preparing the catalogue. If this is the case please specify it in the text. Otherwise update the text with the latest version 7 of the ISC-GEM catalogue.

We were in fact using the most recent version (v. 7) in our comparison, but had not updated the manuscript text. We shall do that, saying "**with version 7 being released in 2020 (Storchak et al., 2013; Di Giacomo et al., 2015)**".

L84: Those were the cut-off magnitude adopted for the first relaese in 2013 (as explained in Storchak et al., 2013; Di Giacomo et al., 2015). However, those cut-off magnitudes were lowered during the extention of the catalogue (see Di Giacomo et al., 2018, doi:10.5194/essd-10-1877-2018). Delete the sentence
Shall be deleted.

L88: This is an abrupt end of the Introduction. There is no clue what the ISC-GEM and Panzera catalogues mentioned here are relevant for in this work? Did the authors use in any way any of them? If not then please clarify why. As it is the text does not give the reader a meaning for last two paragraphs.

We used the former and not the latter. To explain we shall add: "**It is not used as a source for the new catalogue but instead for quality check and comparison**" after the ICS-GEM discussion and "**Unfortunately the IMO magnitudes are very inaccurate, at least when MW ≥ 4 (Fig. 4), and thus this catalogue has not been used directly in the current work**" after the Panzera discussion

In addition, the Introduction should end with a glimpse of what the next sections will cover.
That is correct, a summary was missing. We shall add: "**The next section discusses the primary sources used to compile the new catalogue. This is followed by a two sections describing how epicenters and magnitudes in the catalogue are determined. The final section contains details of the catlogue, including how to retrieve it, as well as a discussion of completeness magnitude, comparison with ISC-GEM, and comparison with the total moment of a simple plate motion model.**"

L98: This sounds a bit measleading. It is true that for one origin time there could from 0 to N magnitudes associated with that origin, but for most of the events M4+ the ISC

Bulletin has multiple origin times (with N magnitudes associated to each origin time). Each event has a prime location (= ISC if the event is relocated by the ISC)

We have looked carefully at the catalogue downloaded from the ISC website cited on line 97 on 23 Nov. 2020, with all available MS, mb and MW values for the NATL region. The shortest time difference for subsequent events in similar location (within 200 km) is 14 seconds, which we deem to represent two different events. We say "For each earthquake a single origin time (UTC) and location **but** multiple magnitude values are provided". We shall change "with" to "but" and hope it maybe less misleading??

L99: It should be explained why the authors do not consider the local magnitude, especially for ICEL region. Their catalogue goes down to M4, and the teleseismic computation of Mw, mb and MS may not provide the best magnitude available for smaller earthquakes (say below 5 or 4.5).

We simply decided that ML is not very standardized... Moreover, the ISC-GEM project does not use ML and we are following their methodology to some extent. In addition many of the ML values are from the REY (Reykjavik) agency, which we are excluding as explained in the manuscript (see also our reply to the L88 comment). We think that these reasons need not be mentioned in the manuscript.

L112: Again, there are 653 GCMT solutions in the ICEL-NMAR region at the time of writing, but we all know very well GCMT now it would have more solutions. Therefore, it would be good to specify when this was done (time of preparing the paper? of the catalogue?).

We shall add the download date (25 Nov. 2020) to the GCMT entry in the reference list cited on line 110. We shall do the same for the ISC and USGS entries.

L113–14: I suggest the authors to review this practice when compiling the catalog. Quoted from the GCMT website (https://www.globalcmt.org/CMTsearch.html):

"The moment magnitude is calculated by this software using the formula of Kanamori (1977), $MW = (2/3)*(\log M0 - 16.1)$, where M0 is given in units of dyne-cm. Prior to February 1, 2006, the quantity $(2/3)*16.1$ was rounded to the value 10.73. For a small number of earthquakes, searches conducted after 2006/02/01 will give values for MW that differ by 0.1 magnitude unit from values given by searches prior to 2006/02/01."

The ISC Bulletin uses the seismic moment from the GCMT solution to obtain Mw to avoid rounding effects, so it may be advisable to stick to the GCMT Mw values listed in the ISC Bulletin.

Currently we say „There are 653 events in the NMAR region in this catalogue, and all but 9 of them are also in the ISC catalogue, marked as originating from GCMT. In 482 cases the MW match but in 171 cases there is a mismatch of 0.1 magnitude, and the average is used here". Unfortunately some of this information is incorrect, it was written before we downloaded the latest versions of the catalogs. We plan to say: „**There are 663 events in the NMAR region in this catalogue, and all but 7 of them are also in the ISC catalogue. The GCMT catalog gives MW with two decimal places, while ISC gives only one, but apart from that most of the values match between the catalogs.**"

Moreover we stopped taking the average and chose instead to use the GCMT value directly, as it provides more decimal places. But that information doesn't belong here, but in section 4, where we say that GCMT MW-values are used verbatim when available.

L113: If I am not mistaken, in the file icel-nmar.txt there are 7 (not 9) GCMT solutions that apparently are not in the ISC catalog:

| 1226 | NMAR | 1982-05-02T07:12:44 | 43.61 | -28.94 | 5.64 0.09 | nan | nan | nan | nan | GCMT | - | - | USGS | GCMT |
| 1266 | NMAR | 1983-01-15T06:43:58 | 73.17 | 5.72 | 5.27 0.09 | nan | nan | nan | nan | GCMT | - | - | USGS | GCMT |
| 1830 | NMAR | 1990-12-04T09:12:51 | 43.76 | -28.87 | 5.51 0.09 | nan | nan | nan | nan | GCMT | - | - | USGS | GCMT |
| 2604 | NMAR | 1999-06-07T16:35:47 | 73.08 | 5.45 | 5.49 0.09 | nan | nan | nan | nan | GCMT | - | - | USGS | GCMT |
| 6543 | NMAR | 2018-05-30T10:30:43 | 76.28 | -1.86 | 5.25 0.09 | nan | nan | nan | nan | GCMT | - | - | USGS | GCMT |
| 6569 | NMAR | 2018-08-09T08:30:32 | 54.25 | -35.25 | 4.78 0.09 | nan | nan | nan | nan | GCMT | - | - | USGS | GCMT |
| 6589 | NMAR | 2018-08-31T15:37:41 | 69.07 | -11.00 | 4.71 0.09 | nan | nan | nan | nan | GCMT | - | - | USGS | GCMT |

However, for all of these events the ISC Bulletin lists the ISC location and ISC magnitudes and includes the GCMT solution, e.g.: http://isc-mirror.iris.washington.edu/cgi-bin/web-db-v4?event_id=598169&out_format=IMS1.0&request=COMPREHENSIVE

We shall correct the 9 to 7; see reply to previous comment. The reviewer is correct that the ISC bullettin does provide the tensor solution when looking up the individual events, but these are not really needed for our catalog, as the GCMT web site provides them in more easily accessible format.

L115: It is indeed a pity that the Ambraseys and Sigbjörnsson catalogue is not available online. It would be good to cross-check the solutions from Ambraseys and Sigbjörnsson with the locations from the ISC-GEM Catalogue. The latter used modern location techiniques and, most likely, a more comprehensive set of stations to recompute locations

We have checked the locations in the ISC-GEM against those in the new catalog within 20 km from the shore of Iceland. We judge our locations to be the most accurate available, with errors of a few kilometers after 1975, and before that the error is unlikely to be more than 10–15 km, at least for large earthquakes as are in the ISC-GEM. There are 18 events after 1975 and the median, 3rd quartile and maximum location difference are 13, 17 and 63 km. Until 1975 there are 15 events with median, 3rd q. and max distance of 22, 53 and 210 km. This difference must be attributed to inaccuracy of the ISC-GEM locations. We have thought about whether to include this information in the manuscript and didn't really find a suitable place to put it, so we plan to leave it out.

L121: please add the reference (BAAS 1913-1917)

BAAS: British Association for the Advancement of Science, Seismological Committee, quarterly issues, 1913–1917.

We shall add this reference

L126: this is not strictly correct, as the USGS lists a preferred magnitude value (often Mww in recent years), but many other magnitudes are also available, e.g.: https://earthquake.usgs.gov/earthquakes/eventpage/us6000afgh/origin/magnitude

We have already addressed this issue and proposed a change in our reply to the comment on line 126 by referee1.

L128: The USGS (reporting under agency NEIC in the ISC Bulletin) is one of the main contributors of the ISC Bulletin. Hence, there is not so much surprise that the "USGS-labeled value" is the same in the ISC Bulletin. However, after the NEIC bulletin is included in the ISC Bulletin, the ISC locations benefits from additional reports and further analyst review. That the ISC and NEIC location differ even of a tens ok km is well-known. What is missing in the sub-section is what use the authors have of the USGS and what matters for this work that USGS locations can differ from ISC locations

The reason that we discuss the USGS catalogue is primarily that we considered it to be the most important international catalogue apart from ISC and GCMT. It it true that it has few magnitudes that are not in ISC, but it is a separate downloadable catalog with independent locations from those of ISC, and the location difference is discussed again and made use of both in section 3.2 and 3.3.

Section 2 discusses all our sources without details of how they are used, and thus we do not see need to add such details for USGS.

L187: please review the use of commas

Two commas will be dropped.

L188–190: this is a risky practice. Particularly in the early part of last century mislocations can be larger than 100 km (and comsidering that many location listed in the catalogue are rounded to the nearest degree), and without station data one may not group origins when necesary and, therefore, create phantom events.

We checked our procedure, and discovered that the referee's concern is valid. We have rerun our earthquake pairing procedure with several combinations of time and distance windows. With 15 s and 300 km we discovered 4 phantom events which we had introduced when combining USGS and ISC: On 1986-06-26 (11 s, 132 km), 1997-12-26 (12 s, 129 km), 2006-06-14 (8 s, 284 km) and 2007-12-10 (4 s, 122 km). In all cases we discovered that the ISC review procedure had decided that these were not separate events. With the same window, when combining GCMT and ISC, 1 phantom event arose, on 2018-08-31 (13 s, 202 km). Finally, when combining AmbSig and ISC we needed to raise the distance window slightly and discovered 1 phantom event on 1964-02-26 (4s, 310 km). If we raised the windows to 25 s and 1000 km (90 s in the AmbSig case) no further misidentified pairings were discovered. We shal change the description in the article and say "**Jones et al. (2000) and several later publications propose that two records that differ by less than 16 s and 100 km refer to the same earthquake. We have discovered that this is too strict, and use windows of 16 s and 320 km. Increasing the window to 25 s and 1000 km gave identical event pairings.**". We shall also change the catalog itself accordingly.

When doing the review of the catalogue we discovered that when available, SIL locations had been selected for all events, even those far offshore Iceland. As these locations become increasingly inaccurate when moving off Iceland, we changed this procedure, and now shall use ISC locations instead for events outside 63°–67°N and 13°–25°W. We shall change and simplify the relevant paragraph on lines 236–242. We remove from "whereas" to the end of the section and insert "**The SIL-locations are however accurate to a few km inside the station network, and they are judged to be more accurate than the teleseismic locations in the region 63°N–67°N and 13°W–25°W**"

After making these changes we have redone the classification of location sources reported in section 3.1, and added similar classification for the period after 1990 at the end of section 3.2. We also redid the counting of events according to magnitude source in section 4.1. All these counts are now done using data available in the final catalogue, instead of the preliminary data file (cf. Section 2.3), and therefore could be verified by the reader. We shall add an explanation of the procedure at the end of section 2.3: "**It** [the data file] **contains some smaller earthquakes that are absent from the final catalogue, as explained at the beginning of Section 4 below. The counts of events according to period, region, location source, and magnitude source, in Sections 3.1, 3.2 and 4.1, are however all made using the catlogue, instead of this data file, as we deem that information to be more relevant for the reader.**"

L194: It seems that the Section regards only ICEL region. Therefore it would be appropriate to specify that in the Section title

We agree and change the title to "**Earthquake locations in the ICEL region**"

L210: more than a relocation this sounds more like fixing the location based on local information

We shall say "and **adjusted the location** of 6 events"

L243: I suggest to rename this sub-section as "Variability of earthquake locations" or simply "Location differences", as to give a measure of location accuracy one should compare earthquake locations with ground-truth locations (hence, I suggest to refrain from using

"accuracy" in the text when referring to earthquake location). Nevertheless, the statistics included in this brief sub-section do not give, strictly speaking, measures of location accuracy but rather statistics on location differences, which, in turn, can be considered as a proxy of the precision of the locations.

We shall rename the section "**Uncertainty of earthquake locations**" and add an explanation at the beginning of the section and say: "**To get some indication of the uncertainty in event locations in the international catalogues we have looked at the variability between diffent catalogs, which can be considered as a proxy of the precision of the locations.** We have also recomputed all the differences and statistics and now include 90th percentiles in addition to medians and maximum.

In addition, it is a bit confusing that the last sentence specifies ISC-USGS comparisons in the ICEL region when ISC-USGS comparison was already mentioned in the previous sentence. Does that refer to the whole ICEL-NMAR then? Please clarify

We are also confused. We shall remove the last comparison.

L252–254: In line with previous comments, I suggest to remove this sentences and start the section with sentence at line 255

We are hesitant to remove the whole sentence, as it explains the methodology underlying the whole project, but we intend to rewrite it and say: "**Contrary to earthquake locations, where local information is crucial, estimating the size of larger earthquakes with teleseismic data is often easier and more reliable than using regional and local data. The dominant periods at teleseismic distances are longer and the structure is smoother due to attenuation of the higher frequencies**"

L256: please review the use of the comma here
OK

L259: [Delete "thus killing two birds with one stone"]
The phrase has already been killed at the request of referee 1 :)

L274: It seems that only 2 Mw are from USGS in the catalogue file:

```
4868  NMAR  2012-05-24T22:47:47  72.96   5.68   6.10 0.11   nan  nan   nan  nan  USGS    -        -        USGS    USGS

6801  NMAR  2019-06-21T06:50:58  47.12  -0.41   4.00  nan   3.85 0.20  4.08 0.24  USGS    CTBTO    Average  ISC     ISC
```

Yes this is correct there are only 2 Mw, and the recounting described at the end of our reply to L188–190 shows that. It will be corrected.

and for event 4868 the GCMT solution is available:

http://isc-mirror.iris.washington.edu/cgi-bin/web-db-v4?event_id=601033410&out_format=IMS1.0&request=COMPREHENSIVE

whereas for event 6801 it is true that Mw comes only from USGS, but one wonders if an event in France should be included in the NMAR region, see also comment for Figure 2

Event 4868 is exactly on the border of the NMAR region, in the primary ISC catalog (not the COMPREHENSIVE) its location is on 73.03°N but in the USGS catalog it is on 72.96°N. This explains the missing GCMT solution. This could of course be corrected by downloading a larger area and make the restriction to the NMAR-region in the final stage. We consider this discrepancy to be not so important to warrant this extra work.

L279: Here the authors may wish to cite a very similar result found by Gasperini et al. (2012), doi:10.1111/j.1365-246X.2012.05575.x
The citation shall be added.

L354: Is there a reason why Mw-sd = nan for 3015 entries in the file icel-nmar.txt ?
These, now 3012, are events with (proxy or modelled) MW < 4.5. All but 52 are modelled,

and one can see from the graphs in Figure 3 that the regression standard error is not well determined in this lower range of the magnitudes. We shall say on line 427 that "**uncertainty was not computed for MW < 4.5 because the regression accuracy is reduced at the lower magnitudes**".

L363:   This regression model was first introduced by Di Giacomo et al. (2015), doi:10.1016/j.pepi.2014.06.005
We shall add a reference to Di Giacomo et al.

L367–368: the histogram equaliztion was suggested by Di Giacomo et al. (2015), doi:10.1016/j.pepi.2014.06.005
We shall add a reference to Di Giacomo et al. also here

L398:   Looking at the proxy Mw values in the catalogue file, one soon notices that the proxy Mw-sd is <= than the sd of its basis, when that is MS. Although this make the proxy uncertainty look good, one would expect that the Mw proxy uncertainty is greater than the uncertainty of its basis. Please clarify why the proxy uncertinaty is smaller than that of its basis
The slope of the MS–MW curve in Figure 3 (top left) is < 1, for MS=4.5 it is 0.64 and for MS=6 it is 0.79. With the chi-square model this slope is a good indicator for the ratio between the sd-estimates for the two magnitude types. We have computed the average sigma-Ms in the final catalogue for (a) $4 < MS < 5$, giving 0.204 and (b) for $5.5 < MS < 6.5$ giving 0.236. Plugging these values into formula (7) using sigma(Mw) = 0.09 gives for the smaller slope:

$$\text{sigma-proxy} = \text{sqrt}(0.64^2 \times 0.204^2 + 0.09^2) = 0.159$$

and for the larger slope:

$$\text{sigma-proxy} = \text{sqrt}(0.79^2 \times 0.236^2 + 0.09^2) = 0.207$$

We also computed the average sigma-Mw for (a) and (b) giving 0.157 and 0.202, so everything fits very well.

We shall add a small explanation: "**Note that because f '(MS) < 1 then sigma-proxy will be smaller than sigma-MS**"

In addition, it appears that in the catalogue file the mb-sd = 0.3 is often listed also for mb = nan, e.g.:

```
0003  ICEL   1905-01-28T06:18:30   63.95 -22.00   5.81 0.21   5.60 0.25   nan nan  proxy  ISCother -   AmbSig  AmbSig
```

Is that on purpose or a feature in writing out the catalogue file?
This was a mistake in the program, which has been corrected.

L406:   Please detail what is MLW or provide a reference for it
The relevant citation shall be added.

L422:   Please make sure that all src fields are correct. For example, the first event (0001) lists MS-src=ISCother, but the event is not in the ISC at all, hence one wonders where ISCother comes from in this case.
We thank the referee for spotting this. This was another mistake in the program which has also been corrected.

Furthermore, for event 0020 the loc-src and time-src are both listed as ISC, but the event does not have an ISC location. In line with the magnitude source nomenclature, the authors should list, in such cases , loc-src and time-src = ISCother
As explained in our reply to the Fig 2 comment the version of the ISC-catalogue that we use contains only one time and location but multiple magnitudes (including the mag source). When we enter ISC as the location source and time source, we are simply reporting where from we obtain the time and location.

L422:  please make sure the references are properly associated in this file. For example, the file reads

> AmbSig    The catalog of Ambraseys and Sigbjornsson [9]

but then in the REFERENCES one finds:

> [9]  International Seismological Centre, 2020. ISC On-line Bulletin. Retrieved 2020-03-31, from http://www.isc.ac.uk/iscbulletin/search/

Also, reference 14 reads:

> "[14] Kristján Jónasson, Bjarni Bessason, Ásdís Helgadóttir, Páll Einarsson, Gunnar B. Gudmundsson, Bryndís Brandsdóttir, Kristín S. Vogfjörd, and Kristín Jónsdóttir. A Harmonized Instrumental Earthquake Catalog for Iceland and the Northern Mid-Atlantic Ridge (the accompanying article). Submitted for publication in Journal of Geophysical Research: Solid Earth in December 2020."

The authors may need to update the journal in [14]
We shall correct these, and try our best to make sure that all associations are correct.

L426:  The authors sometimes use a different source for origin time and location. No explanation about this approach is given. Normally catalogues list an origin time and location from the same source. Mixing origin time and location this way is at least unusual and requires an explanation.
This is a very good remark and we have changed our methodology, and shall now report IMO times when IMO locations are used. The IMO times (if available) will also also used for events with locations taken from published reports, articles and the AmbSig catalog (which rounds times to the nearest minute), as well as events relocated by us. Other times will be taken from the international catalogs. The detailed procedure will be explained just before the references in the supporting info file:

> **The origin time in the catalog is determined with the following rules:**
> **1. If an event is in the Icelandic Meteorological Office (IMO) catalog and its epicenter is not taken from ISC or USGS, then the IMO time is used.**
> **2. Otherwise if the event is in the ISC catalog the ISC time is used**
> **3. Otherwise if the event is in the USGS cataolgue that time is used**
> **4. Otherwise if the event is in the GCMT catalog that time is used**
> **5. Otherwise the event must be in the Ambraseys and Sigbjörnsson catalog and the time is taken from there.**
>
> **The IMO times are probably the most accurate ones, especially after 1990. For completeness, the accomanying file "origin-time.txt" contains all available origin times for each event.**

L427–428: True that depth is the focal paraamter less well constrained in any catalogue, but the studied area has, as the authors point out, quite a well-defined seismogenic thickness. Hence, a field for depth could be added. If no (or not reliable) depth information is available, nan could be used. This should be the case especially for earthquakes in the early part of last century, but in recent years we have a much better idea of the depth of the earthquakes in the study area. Furthermore, for seismic hazard studies depth is relevant.
The main reason for not including hypocentral depth is that the largest earthquakes that are most important for hazard studies rupture the whole seismogenic thickness of the crust, and also, for a large part of the catalogue, the resolution for the depth is low. If we were to include depth it would probably just give the users of the catalogue misleading information.

Note also that there are several references to the hypocentral depth in the comments of referee 1 and our replies to these (lines 31–37, lines 427–429, and supporting info file)